# A HIGH-RESOLUTION AND OBSERVATIONALLY CONSTRAINED OMI NO₂ SATELLITE RETRIEVAL

**Daniel L. Goldberg*[1,2], Lok N. Lamsal [3,4], Christopher P. Loughner [5,6], William H. Swartz[4,7], Zifeng Lu [1,2], and David G. Streets [1,2]**

**[1]Energy Systems Division, Argonne National Laboratory, Argonne, IL 60439, USA**

**[2]Computation Institute, University of Chicago, Chicago, IL 60637, USA**

**[3]Goddard Earth Sciences Technology and Research, Universities Space Research Association, Columbia, MD 21046, USA**

**[4]NASA Goddard Space Flight Center, Code 614, Greenbelt, MD 20771, USA**

**[5]NOAA Air Resources Laboratory, College Park, MD 20740, USA**

**[6]Earth System Science Interdisciplinary Center, University of Maryland, College Park, MD 20740, USA**

**[7]Johns Hopkins University Applied Physics Laboratory, Laurel, MD 20723, USA**

**Paper submitted to**

**Atmospheric Chemistry and Physics**

**Originally submitted: March 9, 2017**
**Revised draft submitted: August 21, 2017**

*Corresponding author. Phone: (630) 252-3931; Fax: (630) 252-8007; Email: dgoldberg@anl.gov.

# A high-resolution and observationally constrained OMI NO$_2$ satellite retrieval

Daniel L. Goldberg[1,2], Lok N. Lamsal[3,4], Christopher P. Loughner[5,6], William H. Swartz[4,7], Zifeng Lu[1,2], and David G. Streets[1,2]

[1]Energy Systems Division, Argonne National Laboratory, Argonne, IL 60439, USA
[2]Computation Institute, University of Chicago, Chicago, IL 60637, USA
[3]Goddard Earth Sciences Technology and Research, Universities Space Research Association, Columbia, MD 21046, USA
[4]NASA Goddard Space Flight Center, Code 614, Greenbelt, MD 20771, USA
[5]NOAA Air Resources Laboratory, College Park, MD 20740, USA
[6]Earth System Science Interdisciplinary Center, University of Maryland, College Park, MD 20740, USA
[7]Johns Hopkins University Applied Physics Laboratory, Laurel, MD 20723, USA

*Correspondence to*: Daniel L. Goldberg (dgoldberg@anl.gov)

**Abstract.** This work presents a new high-resolution NO$_2$ dataset derived from the NASA Ozone Monitoring Instrument (OMI) NO$_2$ version 3.0 retrieval that can be used to estimate surface level concentrations. The standard NASA product uses NO$_2$ vertical profile shape factors from a 1.25° × 1° (~110 × 110 km) resolution Global Model Initiative (GMI) model simulation to calculate air mass factors, a critical value used to determine observed tropospheric NO$_2$ vertical columns. To better estimate vertical profile shape factors, we use a high-resolution (1.33 × 1.33 km) Community Multi-scale Air Quality (CMAQ) model simulation constrained by in situ aircraft observations to re-calculate tropospheric air mass factors and tropospheric NO$_2$ vertical columns during summertime in the eastern United States. In this new product, OMI NO$_2$ tropospheric columns increase by up to 160 % in city centers, and decrease by 20 − 50 % in the rural areas outside of urban areas when compared to the operational NASA product. Our new product shows much better agreement with the Pandora NO$_2$ and Airborne Compact Atmospheric Mapper (ACAM) NO$_2$ spectrometer measurements acquired during the DISCOVER-AQ Maryland field campaign. Furthermore, the correlation between our satellite product and EPA NO$_2$ monitors in urban areas has improved dramatically: $r^2 = 0.60$ in new product, $r^2 = 0.39$ in operational product, signifying that this new product is a better indicator of surface concentrations than the operational product. Our work emphasizes the need to use both high-resolution and high-fidelity models in order to re-calculate satellite data in areas with large spatial heterogeneities in NO$_x$ emissions. Although the current work is focused on the eastern United States, the methodology developed in this work can be applied to other world regions to produce high-quality region-specific NO$_2$ satellite retrievals.

## 1 Introduction

Tropospheric $NO_2$ is a trace gas toxic to human health and during ideal atmospheric conditions can photolyze to create $O_3$ another toxic air pollutant with a longer atmospheric lifetime. The eventual fate of tropospheric $NO_2$ is often $HNO_3$, a chemical species easily dissolved in water and responsible for acid rain. $HNO_3$ can also react with ammonia to create nitrate aerosols, which contribute to haze and are harmful to human health.

There are some natural sources of nitrogen oxides ($NO_x \equiv NO+NO_2$), such as from soil through microbial nitrification and denitrification (Conrad, 1996), lightning (Ridley et al., 1996), and natural wildfires (Val Martin et al., 2006), but the majority of the $NO_2$ in our atmosphere today originates from anthropogenic sources (van Vuuren et al., 2011). When temperatures are greater than 1500 K, such as in fuel combustion, nitrogen ($N_2$) and oxygen ($O_2$) spontaneously react to create NO via the endothermic Zeldovich mechanism. The nitrogen in fuels are also converted to NO during combustion making fuels rich in nitrogen, such as coal, more efficient in creating NO. NO is quickly oxidized to $NO_2$ in the atmosphere, most often by ozone, in a matter of seconds. Thus the NO and $NO_2$ species are often grouped into a single species called $NO_x$. In the presence of hydroperoxy ($HO_2$) or organic peroxy radicals ($RO_2$, where R is any organic group), NO can also be oxidized to $NO_2$ without consuming ozone. This is the rate-limiting step in the chemical chain reaction producing tropospheric ozone.

$NO_2$ has strong absorption features within the 400 – 450 nm wavelength region (Vandaele et al., 1998), which approximately corresponds to violet visible light. Satellite instruments measure the absorption of solar backscatter in the UV-visible spectral range, enabling estimation of the amount of $NO_2$ in the atmosphere between the instrument and the surface. By comparing observed spectra with a reference spectrum, we can derive total column amounts; this technique is called differential optical absorption spectroscopy (DOAS) (Platt, 1994).

$NO_2$ has been continuously measured from satellites for over two decades now. The first instrument to remotely measure $NO_2$ was the Global Ozone Monitoring Experiment (GOME) launched aboard the European Remote Sensing 2 (ERS-2) satellite in April 1995 (Burrows et al., 1999). Despite its coarse temporal and spatial resolution (global coverage once every three days and pixel size of $40 \times 320$ km), it was the first remotely sensed instrument to characterize $NO_2$ columns from space, showing enhanced tropospheric $NO_2$ over North America and Europe (Martin et al., 2002; Martin et al., 2003). In the early 2000s, Scanning Imaging Absorption Spectrometer for Atmospheric Chartography (SCIAMACHY) (Bovensmann et al., 1999) and Ozone Monitoring Instrument (OMI) (Levelt et al., 2006; Bucsela at el., 2006; Boersma et al., 2007) became additional space-based instruments to measure $NO_2$. These instruments were designed to achieve better spatial resolution (SCIAMACHY: $30 \times 60$ km, OMI: $13 \times 24$ km) than GOME. Boersma et al. (2008a) documented the differences between the two retrievals. In early 2012, ground operators lost contact with SCIAMACHY, but OMI is still operational as of 2017. There are two operational OMI $NO_2$ retrievals: the KNMI DOMINO v2.0 product (Boersma et al., 2007) and the NASA OMNO2 v3.0 product (Krotkov et al., 2017).

OMI $NO_2$ has been used to estimate $NO_x$ emissions from various areas around the globe (Streets et al., 2013) including North America (Boersma et al., 2008b; Lu et al., 2015), Asia (Zhang et al., 2008; Han et al, 2015; Kuhlmann et al., 2015), the Middle East (Beirle et al., 2011), and Europe (Huijnen et al., 2010; Curier et al., 2014). It has also been used to generate and validate $NO_x$ emission estimates from source sectors such as soil (Hudman et

al., 2010; Vinken et al., 2014a; Rasool et al., 2016), lightning (Allen et al., 2012; Liaskos et al., 2015; Pickering et al., 2016), power plants (de Foy et al., 2015), aircraft (Pujadas et al., 2011), marine vessels (Vinken et al., 2014b; Boersma et al., 2015), and urban centers (Lu et al., 2015; Canty et al., 2015; Souri et al., 2016). More recently, there has been an emphasis on analyzing emission trends because OMI has been retrieving high-quality tropospheric $NO_2$ data for over ten years. Over this decade, some areas have seen increases, such as India (Lu and Streets, 2012), the

Canadian oil sands region (McLinden et al., 2015), and other oil extraction regions (Duncan et al., 2016), while areas such as the eastern United States (Russell et al., 2012; Lamsal et al., 2015; Krotkov et al., 2016; de Foy et al., 2016a) and Europe (Curier et al., 2014; Duncan et al., 2016) have seen large decreases due to a switch to cleaner fuels and the implementation of emission control technologies. Over this 10-year period, China has seen a reversal of its trends: during 2005-2010 OMI $NO_2$ tropospheric columns were increasing (Verstraeten et al., 2015), in 2011-

2012 they had stabilized (Souri et al., 2017), and since 2012 they have subsequently decreased as the country enforces its Twelfth 5-year plan (de Foy et al., 2016b).

Remote sensing instruments typically measure the entire column content instead of in situ concentrations at individual vertical levels. Being able to derive surface concentrations from column content information would be very useful for the policy-making and health-assessment communities. In particular, detecting the spatial

heterogeneities of $NO_2$ in and around city centers are of strong interest as many people are exposed to $NO_2$ or co-located pollutants exceeding policy thresholds in these areas. Satellite measurements with spatial resolution > 13 km, such as OMI, have difficulty observing the fine structure of $NO_2$ plumes at or near the surface (e.g., highways, power plants, factories, etc.) (Chen et al., 2009; Ma et al., 2013; Flynn et al., 2014), which are often less than 10 km in width (Heue et al., 2008). This can lead to a spatial smoothing of pollution, which does not exist in reality

(Hilboll et al., 2013). Remote sensing instruments with finer spatial resolution, such as TROPOMI (Veefkind et al., 2012) and TEMPO (Zoogman et al., 2017), may be able to resolve this issue.

Until the next generation of satellites is launched, there have been several techniques to modify OMI $NO_2$ data a posteriori. Kim et al. (2016) developed a technique in which users can utilize regional air quality model information to spatially downscale OMI $NO_2$ measurements. This technique has shown to increase the variability of OMI $NO_2$

within urban areas, which is in better agreement with observations in these regions. In another effort to merge model and satellite data, Lamsal et al. (2008) was able to infer surface level $NO_2$ concentrations from OMI $NO_2$ by applying local scaling factors from a global model. There has also been an emergence of a technique that combines land-use regression techniques with satellite information to infer ground-level $NO_2$ concentrations (Novotny et al., 2011; Vienneau et al., 2013; Lee et al., 2014; Bechle et al., 2015; Young et al., 2016). While each individual

technique is useful, all of the aforementioned techniques use model data to adjust existing satellite data, but do not address issues inherent with the satellite retrieval methodology.

Previous studies have shown that the air mass factor, a value needed to convert the slant column measurement into a vertical column amount, is one of the largest source of uncertainty in the OMI $NO_2$ retrieval, contributing up to half of the total error (Boersma et al., 2004; Lorente et al., 2017). There are two existing OMI $NO_2$ products that use information from a regional chemical transport model to re-calculate the air mass factor: Berkeley High-Resolution (BeHR) $NO_2$ (Russell et al., 2011; Laughner et al., 2016) and City University of Hong Kong OMI (HKOMI) $NO_2$ (Kuhlmann et al., 2015). BeHR $NO_2$ uses a monthly averaged $12 \times 12$ km Weather Research and Forecasting coupled with Chemistry (WRF-Chem) model simulation with higher resolution terrain pressure and Moderate-resolution Imaging Spectroradiometer (MODIS) black-sky albedo to re-calculate the air-mass factors for the United States. This study found that by updating the air mass factors with a high-resolution simulation, $NO_2$ tropospheric vertical columns increased in urban areas and decreased in rural areas when compared to typical satellite products processed with global model simulations (Russell et al., 2011). More recently, the BeHR $NO_2$ product has been updated (summer 2013 only) to account for daily variations in shape profiles and terrain pressure, which modifies daily retrievals by as much as 40% (Laughner et al., 2016). The HKOMI product uses $NO_2$ shape profile, terrain elevation, and meteorological information from a $3 \times 3$ km a Community Multiscale Air Quality (CMAQ) simulation coupled offline to a Weather Research and Forecasting (WRF) simulation to re-calculate the air mass factor for the Pearl River Delta region of China. Similarly, they found that the tropospheric vertical $NO_2$ columns increased in an urban area; this improved agreement between satellite and ground observations (Kuhlmann et al., 2015). One critical limitation of the BeHR and HKOMI products is the lack of lightning $NO_x$ emissions in the model simulations used to derive the air mass factor. The POMINO product takes a slightly different approach. This study improves the air mass factor for China (Lin et al., 2015) by (1) using improved information on surface albedo (MODIS Bidirectional Reflectance Distribution Function (BRDF)), (2) improving the treatment of aerosols and cloud pressure/fraction, and (3) using a nested ($0.667° \times 0.5°$) GEOS-Chem simulation for the $NO_2$ shape profiles. These three changes increase annual mean $NO_2$ tropospheric vertical columns by $15 - 40\%$. A summary of all available OMI $NO_2$ retrievals are listed in Table 1.

We build upon these studies by using an even higher resolution (1.33 km) regional air quality model to generate air mass factors for urban metropolitan areas in the mid-Atlantic region of the eastern United States. Use of such resolution allows calculation of air mass factors representing OMI ground pixels. The new air mass factors are then used to re-calculate $NO_2$ tropospheric vertical columns. Furthermore we utilize a technique for constraining the $NO_2$ shape profiles to aircraft observations and invoke a new downscaling method developed by Kim et al., (2016) to enhance the content of the satellite observations.

**2 Methods**

**2.1 OMI NO₂**

The Ozone Monitoring Instrument (OMI) has been operational on NASA's Earth Observing System (EOS) Aura satellite since October 2004 (Levelt et al., 2006). The satellite follows a sun-synchronous, low-earth (705 km) orbit with an equator overpass time of approximately 13:45 local time. OMI measures total column amounts in a 2600

km swath divided into 60 unequal area "field-of-views", or pixels. At nadir (center of the swath), pixel size is $13 \times 24$ km, but at the swath edges, pixels can be as large as $26 \times 128$ km. In a single orbit, OMI measures approximately 1650 swaths and achieves daily global coverage over $14 - 15$ orbits (99 minutes per orbit). OMI measures solar backscatter within the 270-500 nm wavelength range. For this paper, we focus on the $NO_2$ retrieval which is derived from measurements in the $400 - 450$ nm range. Since June 2007, there has been a partial blockage of the detector's full field of view, which has limited the number of valid measurements; this is known in the community as the row anomaly: http://projects.knmi.nl/omi/research/product/rowanomaly-background.php.

OMI measures radiance data between the instrument's detector and the Earth's surface. Comparison of these measurements with a reference spectrum (i.e., DOAS technique), allows for calculation of the total slant column density (SCD), which represents the integrated $NO_2$ abundance from the sun to the surface, through the atmosphere, to the instrument's detector. For tropospheric air quality studies, vertical column density (VCD) $NO_2$ data are more appropriate. This is done by subtracting the stratospheric slant column from the total (tropospheric + stratospheric) slant column and dividing by the tropospheric air mass factor (AMF), which is defined as the ratio of the SCD to the VCD, as shown in Eq. (1):

$$VCD_{trop} = \frac{SCD_{total} - SCD_{strat}}{AMF_{trop}} \quad , \text{ where } AMF_{trop} = \frac{SCD_{trop}}{VCD_{trop}} \tag{1}$$

The tropospheric AMF has been derived to be a function of the optical atmospheric/surface properties (surface albedo, aerosols, cloud fraction, and cloud height) and a priori $NO_2$ shape profile (Palmer et al., 2001; Martin et al., 2002) and can be calculated as follows (Lamsal et al., 2014) in Eq. (2):

$$AMF_{trop} = \frac{\Sigma_{surface}^{tropopause} SW \times x_a}{\Sigma_{surface}^{tropopause} x_a} \tag{2}$$

Where $x_a$ is the partial column $NO_2$. The optical atmospheric/surface properties are characterized by the scattering weight (SW) and are calculated by a forward radiative transfer model (TOMRAD), which are output as a look-up table (LUT). The SWs are then adjusted real-time depending on observed viewing angles, surface albedo, cloud fraction, and cloud pressure. For this study, we follow previous studies (e.g., Palmer et al., 2001, Martin et al., 2002, Boersma et al., 2011, Bucsela et al., 2013) and assume that SWs and $NO_2$ profile shapes are independent. The a priori $NO_2$ profile shapes ($x_a$) must be provided by a model simulation. In an operational setting, NASA uses a monthly-averaged and year-specific Global Model Initiative (GMI) model (1.25° lon × 1° lat; ~110 km × 110 km in the mid-latitudes) simulation to provide the a priori $NO_2$ shape profiles. For this study, we derive tropospheric VCDs using a priori $NO_2$ shape profiles from a regional CMAQ simulation. A description of this methodology is included in Section 2.5. All other parameters from the NASA Level 2 product including the total SCD, the stratospheric SCD (which is inferred using a local analysis of the stratospheric field (Bucsela et al., 2013)), the OMI $O_2$-$O_2$ cloud pressure/fraction algorithm (which uses a look-up table to convert $O_2$-$O_2$ column density and continuum reflectance into cloud pressure/fraction (Acarreta et al., 2004)), the surface albedo (which is derived from OMI Lambert Equivalent Reflectance (LER) (Kleipool et al., 2008)), and the SW remain unchanged.

We filter the Level 2 OMI $NO_2$ data to ensure only valid pixels are used. Daily pixels with solar zenith angles $\geq$ 80°, cloud radiance fractions $\geq$ 0.5, or surface albedo $\geq$ 0.3 are removed as well as the five largest pixels at the swath edges (i.e., pixel numbers $1 - 5$ and $56 - 60$). Finally, we remove any pixel flagged by NASA including pixels with NaN values, 'XTrackQualityFlags' $\neq$ 0 or 255 (RA flag), or 'VcdQualityFlags' $>$ 0 and least significant bit $\neq$ 0

(ground pixel flag).

**2.2 DISCOVER-AQ $NO_2$ observations**

In the validation of our new satellite product, we use in situ $NO_2$ observations from the DISCOVER-AQ Maryland field campaign. DISCOVER-AQ was a four-part field experiment designed to probe the atmosphere near urban areas in excruciating detail from aircrafts, ground station networks, and satellites. The first experimental campaign

took place in Maryland (Baltimore, MD - Washington D.C. area) in July 2011. This campaign was particularly unique for an aircraft field campaign in that the focus was limited to single metropolitan area, whereas in other aircraft campaigns, spatial coverage is often over a larger domain. We utilize data acquired by four sources during this campaign: the P3-B aircraft, the ground-based Pandora spectrometer network, the Airborne Compact Atmospheric Mapper on the UC-12 aircraft, and the long-term EPA ground monitoring network. A typical P3-B

aircraft and UC-12 flight path, Pandora $NO_2$ spectrometer locations, and ground monitor locations are shown in Figure 1. DISCOVER-AQ observations were retrieved from the online data archive: http://www-air.larc.nasa.gov/cgi-bin/ArcView/discover-aq.dc-2011. A further description of DISCOVER-AQ Maryland can be found in Crawford et al. (2014).

**2.2.1 P3-B aircraft data**

We use P3-B aircraft $NO_2$ data gathered by the Cohen group (instrument reference: (Day et al, 2002)) to assess the accuracy of our model simulation. This instrument does not have the same positive bias as chemiluminescence $NO_2$ detectors, so there is no need to modify $NO_2$ concentrations by applying an empirical equation (e.g., Lamsal et al., 2008). We utilize one-minute averaged P3-B data from all fourteen flights during July 2011. One-minute averaged data is already pre-generated in the data archive. Hourly output from our model simulation is spatially and

temporally matched to the observations. We then bin the data into different altitude ranges for our comparison.

**2.2.2 Pandora $NO_2$ data**

Measurements of total column $NO_2$ from the Pandora spectrometer (instrument reference: (Herman et al., 2009)) are used to evaluate the OMI $NO_2$ satellite products. Valid OMI $NO_2$ pixels are matched spatially and temporally to Pandora total column $NO_2$ observations. To smooth the data and eliminate brief small-scale plumes or anomalies,

we average the Pandora observations over a two hour period ($\pm$ one hour of the overpass time) before matching to the OMI $NO_2$ data. During July 2011, there were twelve Pandora $NO_2$ spectrometers operating during the experiment; this corresponded to only seventy-nine instances in which valid Pandora $NO_2$ observations matched valid OMI $NO_2$ column data.

### 2.2.3 Airborne Compact Atmospheric Mapper (ACAM) NO₂ data

The UC-12 aircraft was outfitted with a downward looking spectrometer called the Airbone Compact Atmospheric Mapper (ACAM) during the DISCOVER-AQ Maryland campaign (instrument reference: (Kowalewski and Janz, 2009)). The instrument collects hyperspectral measurements in the UV, visible, and near-infrared range from an altitude of approximately 8 km. From these measurements, tropospheric column $NO_2$ underneath the aircraft can be calculated (Lamsal et al., 2017). An ACAM pixel is considered valid, if there are no clouds between the instrument's detector and the surface. Valid OMI $NO_2$ pixels are matched spatially and temporally (± one hour of the satellite overpass time) to the ACAM column $NO_2$ observations. During July 2011, there were only six days in which the UC-12 flight paths overlapped an OMI $NO_2$ swath; this corresponded to only 107 OMI $NO_2$ pixels which could be compared to the ACAM $NO_2$.

### 2.2.4 EPA ground monitor data

There are eighteen EPA $NO_2$ monitoring sites within our study area that were operational during the 5-year period of interest. We gathered this data from the EPA AQS Data Mart (EPA, 2016). Monitoring data were filtered so that only days with valid satellite data were included. To smooth the data, we average all valid ground observations between 12 – 4 PM local time. All EPA monitors measure $NO_2$ by the chemiluminescence method which has a high bias when compared to other techniques (Dunlea et al., 2007; Lamsal et al., 2008; Lamsal et al., 2015). Dunlea et al. (2007) has shown the high bias to be 22 % in a polluted urban environment and as large as 50 % during the mid-afternoon. Lamsal et al. (2008) suggests the bias may be even higher, 50 – 65 %, in the eastern U.S. during the summertime. For this reason, we refer to $NO_2$ from these monitors as $NO_2$*.

### 2.3 GMI model simulation

The operational NASA OMI $NO_2$ product uses a Global Modeling Initiative (GMI) (Strahan et al., 2007) model simulation with a horizontal resolution of 1° × 1.25° (~110 × 110 km) sampled at the OMI overpass time to calculate a priori $NO_2$ shape factors. The model is driven by assimilated meteorological fields from the Goddard Earth Observing System (GEOS) at the NASA Global Modeling and Assimilation Office (GMAO, ttp://gmao.gsfc.nasa.gov/). The GEOS-5 meteorological data are provided every 3–6 h (3 h for surface fields and mixing depths) at 72 pressure levels in the vertical, extending from surface to 0.01 hPa. The model includes the latest available inventories for anthropogenic emissions as discussed in Strode et al. (2015) and Krotkov et al. (2017). These emissions are updated annually with annual scale factor estimates provided by individual countries (van Donkelaar et al., 2008). The GMI model also includes $NO_x$ emissions from soil, lightning, biomass burning, biofuel, and aircraft sources, as described in Duncan et al. (2007) with updates as discussed in Krotkov et al. (2017). The GMI simulation is conducted for 2004-2014, sampling hourly model output at the OMI overpass time. The standard operational retrieval is based on yearly-varying monthly average $NO_2$ profiles derived from the GMI simulation.

### 2.4 CMAQ model simulation

For the high-resolution OMI $NO_2$ product, we use a CMAQ regional model simulation initially prepared for use in Loughner et al. (2014). CMAQ v5.0 is driven off-line by meteorological inputs from the WRF model v3.3 for June and July 2011. Horizontal spatial resolution of both WRF and CMAQ is at 1.33 km. Both models also include 34 vertical levels between the surface and 100 hPa, with 16 layers within the lowest 2 km. The ACM2 drives the boundary layer parametrization in WRF, while ACM computes the convective mixing in CMAQ. Anthropogenic emissions are projected to 2012 from the 2005 EPA National Emissions Inventory (NEI); the 2011 NEI was unavailable when this model simulation was originally completed. Biogenic and lightning emissions are calculated online; biogenic emissions are calculated using BEIS v3.14. Soil $NO_x$ emissions are not included here because the CMAQ soil $NO_x$ parametrization was implemented in a newer version of the model (Rasool et al., 2016). This model simulation utilizes CB05 gas-phase chemistry. The 1.33 km simulation, which we use exclusively in this study, is nested inside three larger domains: 36 km, 12 km, and 4 km. Boundary conditions for the 36 km domain are provided by the MOZART-4 global model. The top of the model assumes "zero gradient", which means the top boundary has concentrations equal to the top model layer. The CMAQ 1.33 km model domain is shown in Figure 1. For additional details, including a discussion on the uncertainty of the meteorological and chemical fields in this simulation, please reference Loughner et al. (2014). Our study is particularly unique in that we use a 1.33 km simulation in lieu of a model with a horizontal resolution more typical of OMI (>13 km). We do this so that we can capture the fine-scale variability within urban areas that cannot be simulated by coarser models and observations.

### 2.5 Air Mass Factor Re-Calculation using CMAQ

To re-calculate the air mass factor for each OMI pixel, we first compute interim air mass factors for each CMAQ model grid cell. The interim air mass factor for each CMAQ grid cell is a function of the $NO_2$ shape factor from the model grid cell and scattering weight from the OMI pixel that overlaps it. We then average all interim air mass factors within an OMI pixel (usually 100's) to generate a single tropospheric air mass factor for each individual OMI pixel. This new air mass factor is used to convert the total slant column into a tropospheric vertical column using Equation 1. Model outputs were sampled at the local time of OMI overpass. For June and July 2011, we use daily $NO_2$ profiles and terrain pressures (e.g., (Laughner et al., 2016)) to re-calculate the AMF. For years other than 2011, we used 2011 monthly mean values of $NO_2$, temperature, and tropopause pressures for the calculation of the AMF. Once the tropospheric vertical column of each OMI pixel was re-calculated, the product was oversampled for June and July over a 5-year period (2008-2012; 10 months total).

### 3 Results

In this section, we describe the new high-resolution satellite product and our validation efforts. Unless otherwise noted, all OMI $NO_2$ results presented here are vertical column densities. First, we compare a priori $NO_2$ shape profiles simulated by GMI (global model) and CMAQ (regional model). Next we develop an initial OMI $NO_2$ satellite product (OMI_CMAQ) using AMFs generated from the CMAQ a priori $NO_2$ profiles. We introduce two

additional steps: improving a priori $NO_2$ shape profiles using aircraft observations and applying a spatial weighting kernel to further improve the spatial distribution of $NO_2$. We then evaluate our new product by comparing to DISCOVER-AQ observations. And finally, we compare the new OMI $NO_2$ product with $NO_2$ VCDs from the original CMAQ simulation.

**3.1 Evaluating modeled $NO_2$ shape profiles: GMI vs. CMAQ**

Trace gas shape profiles provided by model simulations are a critical input to satellite retrievals. To understand the effects of model choice on the a priori $NO_2$ shape profile, we compare the mean 2 PM local time tropospheric $NO_2$ vertical profiles from CMAQ and GMI at several locations in the mid-Atlantic during June & July 2011. In the left panels of Figure 2, we show the mean $NO_2$ mixing ratio as function of altitude for three locations: downtown Baltimore Maryland (an urban area), the Morgantown Power Plant located in Newburg, Maryland 60 km south of the District of Columbia (D.C.), and Arendtsville, Pennsylvania (rural), a location 100 km northwest of Baltimore and upwind of major metropolitan areas during days with climatologically westerly winds. All three locations are shown on Figure 1. In Baltimore, GMI simulates a mean 2 PM surface $NO_2$ mixing ratio of 2.2 ppbv, while CMAQ simulates 9.6 ppbv at the same location. The "Oldtown" monitoring site in Baltimore registered a surface $NO_2*$ mixing ratio of 10.5 ppbv within +/- 2 hours of valid co-located satellite overpasses. As discussed in Sect. 2.2.4, the corrected surface $NO_2$ mixing ratio is approximately 22% lower (but may be up to 65% lower) than observed $NO_2*$; our best surface estimate of 8.2 ppbv with error bars [3.7, 10.5] is shown by the black triangle on Figure 2. The surface value simulated by CMAQ (9.6 ppbv) is much closer to the observed value than GMI (2.2 ppbv). In the second row of panels, the panels representing the Morgantown power plant, CMAQ simulates a plume of $NO_2$ above the surface; the max value is 11.8 ppb corresponding to an altitude of 460 m. The GMI simulation cannot resolve power plant plumes. This yields significant errors in the $NO_2$ shape profiles simulated by GMI near observed large point sources. In the bottom row of panels, we show a location in rural Pennsylvania. CMAQ, once again, does better in simulating the surface concentration than GMI.

However, in the free troposphere (i.e., above 3 km and below the tropopause) CMAQ consistently simulates smaller $NO_2$ than GMI. CMAQ simulates $NO_2$ mixing ratios between 0.01- 0.04 ppbv, while GMI simulates $NO_2$ mixing ratios between 0.06 – 0.09 ppbv over the same altitudes; GMI simulates values which are a factor of three higher than CMAQ. To determine whether lightning NO is the primary driver of this difference, we compare lightning NO emissions from both model simulations in Figure 3. The CMAQ model ingests lightning NO emissions that are an order of magnitude larger than the GMI simulation at most altitudes. This is likely due to WRF simulating more convective precipitation and higher cloud-top heights, both input variables to the lightning NO parameterization, than GMI. Therefore, the differences in free tropospheric $NO_2$ between the two models do not arise from the lightning $NO_x$ parameterizations, but instead from a combination of the chemistry, aviation emissions, vertical mixing, long-range transport, and stratospheric-tropospheric exchange.

### 3.2 Calculation of air mass factors: GMI vs. CMAQ

A normalization of the $NO_2$ as a function of altitude (i.e., $x_a / \Sigma x_a$ in Eq. (2)) is the next step in the calculation of the AMF; these values are defined in the literature as shape factors. The center column panels show $NO_2$ shape factors for three locations. In Figure 2b (Baltimore), the GMI and CMAQ shape profiles (i.e., shape factors as a function of altitude) appear to be similar, but there are noticeable differences within the boundary layer and free troposphere. In Figure 2e, CMAQ captures a localized power plant plume, while GMI does not; this yields large differences in the shape profile within the boundary layer. And in Figure 2h, CMAQ suggests that the $NO_2$ gradient near the surface is not as sharp.

Since the AMF is also a function of the SW, small differences in $NO_2$ shape profiles can manifest very different AMFs. For example, small differences in the shape profile at 7.5 km, where the SW is a maximum (SW = 2.9), have an order of magnitude larger effect than differences at the surface (SW = 0.4).

To fully understand the differences caused by the new $NO_2$ shape factors, we multiply the two shape factors by the satellite scattering weights. Here we define the shape factors × scattering weight ((i.e., $(x_a \times SW)/ \Sigma x_a$ in Eq. (2)) as the adjusted shape factors. This is analogous to the values used for calculation of the air mass factor. The AMF is the integral of the adjusted shape factors with respect to height. In Figure 2c, the CMAQ adjusted shape profile shows values much closer to zero above 3 km than GMI. By using a priori shape profiles from CMAQ, we are enhancing the sensitivity of satellite observations to $NO_2$ concentrations within the boundary layer in Baltimore. In Figure 2f, the adjusted shape profiles are even more dramatic. At this location, adjusted shape profile values from CMAQ are relatively large below 1 km, and close to zero above 1 km, while GMI shows nearly an order of magnitude larger sensitivity above 1 km. In Figure 2i, CMAQ shows larger values above the surface, but within the boundary layer, while GMI shows larger values directly at the surface. In areas, such as these, the adjusted shape factors yield only small changes. In Figures 2c and 2f, the area underneath the red curve is smaller than the area underneath the blue curve. This will yield smaller AMFs when using CMAQ at these locations. As a result, we should expect the new OMI tropospheric $NO_2$ columns to be larger near urban areas and point sources which cannot be resolved by global models. At the rural location, the areas underneath the two curves are roughly the same, yielding similar AMFs and $NO_2$ columns.

### 3.3 Calculation of OMI tropospheric column NO₂

#### 3.3.1 Using CMAQ profiles

We use the AMFs based off the CMAQ simulation to generate $NO_2$ tropospheric column amounts; we call this the OMI_CMAQ product. For this product, tropospheric $NO_2$ columns are calculated from the NASA Level 2 OMI $NO_2$ total slant column using Eq. (1). For AMFs calculated in the months of June and July 2011, we use AMFs derived from daily $NO_2$ shape factors as described by Laughner et al. (2016), resulting in more day-to-day variability in the AMF. Daily CMAQ $NO_2$ shape profiles from the hourly output are matched temporally and

spatially to the OMI pixel. For years other than 2011, we use a two-month (June and July) average of the 2011 $NO_2$ shape factors to derive "summertime" AMFs. Since the resolution of CMAQ is finer than the resolution of OMI, we average all CMAQ AMFs across each individual pixel. Often there are over two-hundred CMAQ AMFs within a single pixel. Since CMAQ is capturing the spatial heterogeneities in urban areas, using it in lieu of GMI to provide

$NO_2$ shape profiles can yield large variability in the AMF between adjacent OMI pixels.

Figures 4a and 4b depict the OMI $NO_2$ tropospheric columns using a priori shape profile information from GMI (OMI_GMI; Figure 4a) and CMAQ (OMI_CMAQ; Figure 4b) in calculating the AMF. Both products are oversampled to 1.33 km for June & July over a 5-year period (2008-2012) by re-gridding to the CMAQ model grid and then averaging the data over the 10-month (two months × five years) period. We have chosen the June & July

timeframe because the CMAQ simulation is only available during these two months. For the OMI_GMI product, the tropospheric $NO_2$ columns were taken directly from the NASA OMI $NO_2$ v3.0 Level 2 product. Figure 5a shows the ratio between the two products.

In the new product (OMI_CMAQ), there are large increases of the $NO_2$ VCDs in city centers. In the operational OMI_GMI product, over the 5-year period, the maximum tropospheric $NO_2$ column within Baltimore city limits is

$3.9 \times 10^{15}$ molecules per $cm^2$. By contrast, in the OMI_CMAQ product, the maximum tropospheric $NO_2$ column within Baltimore city limits is $7.2 \times 10^{15}$ molecules per $cm^2$. These results indicate that by using a regional model, the tropospheric $NO_2$ VCDs in urban areas rise dramatically. This is due, in part, to the regional model being able to better capture $NO_2$ concentrations in the lower-most part of the troposphere (i.e., Figure 2). In suburban and rural locations, $NO_2$ tropospheric VCDs are roughly the same. For example, at the rural Pennsylvania (Arendtsville)

location, the $NO_2$ tropospheric column in the operational product is $2.8 \times 10^{15}$ molecules per $cm^2$ and $2.7 \times 10^{15}$ molecules per $cm^2$ in the new OMI_CMAQ product.

### 3.3.2 Improving modeled vertical profile information with in situ observations

While using CMAQ to calculate AMFs yields a marked improvement in simulating profile shape when compared to using GMI, this CMAQ simulation has a high bias in the calculation of total reactive nitrogen oxides ($NO_y$)

(Goldberg et al., 2014; Anderson et al., 2014), which must be accounted for. Many literature sources, including others using different model set-ups (all are based on the NEI), also show a high bias in simulating summertime column $NO_2$ (Canty et al., 2015; Souri et al., 2016), $NO_x$ (Travis et al., 2016), and $NO_y$ (Goldberg et al., 2016).

In Figure 6, we show $NO_2$ observations acquired by the P3-B aircraft in the early afternoon between 300 m and 3 km during DISCOVER-AQ Maryland matched to CMAQ and GMI output. $NO_2$ mixing ratios simulated by CMAQ

are consistently larger throughout the mid- and upper-boundary layer and lower free troposphere ($1 - 3$ km) by up to a factor of three, but there is fairly good agreement below 1 km; similar results were found by Flynn et al. (2016). The $NO_2$ mixing ratios simulated by GMI below 1 km are a factor of two lower than the P3-B observations. Furthermore, the variability is an order of magnitude smaller than the observations. These shortcomings of GMI are a result of using a monthly mean (the same value used for the satellite retrieval) and coarse resolution model.

Since the P3-B aircraft has limited measurements above 3 km, we have to use estimates from other literature sources to determine the validity of CMAQ in the free troposphere. Lamsal et al. (2017) used measurements from the Airborne Compact Atmospheric Mapper (ACAM) to deduce that GMI is better than CMAQ at simulating $NO_2$ in the free troposphere. In the upper free troposphere, above 10.5 km, Travis et al. (2016) note that $NO_2$ is significantly underestimated by global models, such as GMI. As shown in Figure 2, CMAQ simulates even lower $NO_2$ concentrations than GMI at these altitudes.

We apply a scaling factor inferred from in situ aircraft observations to account for the high model bias below 3 km, and low model bias above 3 km; this is a simplified form of data assimilation. Below 3 km, the model was scaled to observations from the P3-B by multiplying the original values at these altitudes by the fraction of $NO_2$ actually observed. For example, modeled $NO_2$ between 1000 – 1500 m was multiplied by 0.63 to account for the model high bias within this altitude bin. This procedure was repeated for all altitude bins in 500-m intervals from the surface up to 3 km. It should be noted that aircraft measurements from the DISCOVER-AQ Maryland campaign took place only within the Baltimore metropolitan region, and thus these scaling factors may not be fully applicable to upwind rural regions, and certainly cannot be applied to locations outside the eastern United States. Between the altitudes of 3 km – 10.5 km, we switched out the $NO_2$ mixing ratios from CMAQ for $NO_2$ mixing ratios from GMI. Between 10.5 km and the tropopause, we use GMI $NO_2$ mixing ratios multiplied by a factor of three; this scaling factor is based on summertime $NO_2$ observations during the SEAC$^4$RS field campaign as described by Travis et al. (2016).

Using these scaled mixing ratios, we then re-calculate the AMFs and corresponding tropospheric $NO_2$ columns. Figure 4c shows observationally-constrained OMI_CMAQ (OMI_CMAQ_O) tropospheric $NO_2$ columns during the same 5-year summertime period. $NO_2$ tropospheric columns in this product are smaller in magnitude than OMI_CMAQ, and yet still noticeably larger in urban areas than the operational OMI_GMI retrieval (i.e., in Baltimore OMI_GMI: $3.9 \times 10^{15}$ , OMI_CMAQ: $7.2 \times 10^{15}$, OMI_CMAQ_O: $5.0 \times 10^{15}$). Retrievals in upwind rural areas in this new product are now lower than the operational product (i.e., in Arendtsville OMI_GMI: $2.8 \times 10^{15}$ , OMI_CMAQ: $2.7 \times 10^{15}$, OMI_CMAQ_O: $1.7 \times 10^{15}$).

The large reduction in $NO_2$ tropospheric columns between OMI_CMAQ and OMI_CMAQ_O is an outcome of using larger AMFs. The larger AMFs are a result of the original overestimate within the boundary layer and underestimate in the free troposphere. This is a particularly important finding because it means that a model with large biases in the simulation of $NO_2$ can yield poor tropospheric vertical column contents, despite high spatial resolution. This emphasizes the need to evaluate the emissions and chemistry of a model before it is used for satellite retrievals.

### 3.3.3 Enhancing spatial resolution with spatial weighting kernels

Finally in a last step, we apply the method described by Kim et al. (2016) to downscale the OMI retrieval. This method applies a spatial-weighting kernel to portions of each pixel based on the estimated influence from each locality within the pixel. For example, if one side of a pixel overlaps a polluted region, while the other side of the

pixel overlaps a cleaner area, the operational OMI product will denote that the entire area is moderately polluted. Instead, we weight portions of the individual pixel based on the variability simulated in CMAQ. Using this method, the quantity of the satellite data is numerically preserved. This yields a higher resolution snapshot of tropospheric column $NO_2$ that is still constrained by satellite data. Please reference Kim et al. (2016) for a visual representation

of this method.

We call this product OMI_CMAQ observationally-constrained + downscaled (OMI_CMAQ_OD). Figure 4d shows OMI_CMAQ_OD tropospheric $NO_2$ columns. There is now large variability throughout the region, which is typical of a pollutant with a short lifetime ($< 1$ day) such as $NO_2$ in the summertime. $NO_2$ tropospheric columns in urban cores are now significantly larger than the operational product, (i.e., in Baltimore OMI_GMI: $3.9 \times 10^{15}$,

OMI_CMAQ_OD: $10.2 \times 10^{15}$). The largest increases occur near power plants, cement kilns, and major highways. OMI_CMAQ_OD in upwind rural areas are $20 - 50$ % lower than the operational product (i.e., in Arendtsville OMI_GMI: $2.8 \times 10^{15}$, OMI_CMAQ_OD: $1.6 \times 10^{15}$).

While this new product shows power plant plumes that are two to three times larger, we are not suggesting that emissions from power plants are larger than we thought. Instead we are suggesting that the spatially downscaled

OMI product can now "see" these individual plumes, whereas in the operational product, these plumes are blended into an average across the entire OMI pixel. In rare cases, oversampling the operational product in and around very large rural point sources, can denote large power plant plumes (deFoy et al., 2015), but up until this point, smaller point sources or localized sources near major urban areas could not be seen in an OMI $NO_2$ product.

**3.4 Comparison of satellite products to in situ observations**

**3.4.1 Comparison to the Pandora $NO_2$ spectrometer network**

During DISCOVER-AQ Maryland, total column $NO_2$ was measured by a network of twelve Pandora instruments (Herman et al., 2009). We match daily valid Pandora $NO_2$ and valid satellite overpass information, and plot the information in Figure 7a. To calculate total OMI columns, we add the vertical stratospheric column information, a variable in the NASA OMI $NO_2$ Level 2 files, to the OMI tropospheric retrievals. While the operational product

(OMI_GMI) shows some agreement at low values, it has poor agreement when observed $NO_2$ column amounts are greater than $10 \times 10^{15}$ cm$^{-2}$. This is due to coarse resolution of OMI pixels ($24 \times 13$ km at nadir) and the AMFs computed with GMI a priori $NO_2$ profiles, among potential other factors. The slope of the OMI_GMI best-fit line is 0.44, representing a striking low bias at high values, and the $r^2 = 0.10$ denoting almost no correlation; similar results were found by Ialongo et al. (2016).

Table 2 shows the statistical comparison between the satellite products and observations. All OMI_CMAQ products yield slopes closer to one indicating that they are better at capturing the variability between low and high values observed by the ground monitors. The OMI_CMAQ_OD product eliminates the bias altogether. The slope of the OMI_CMAQ_OD best-fit is 0.99 and the $r^2$ increases. An improved but still low $r^2$-value in the newest product may

indicate that a 1.33 km CMAQ simulation provides an improvement, but not an identical match, of daily $NO_x$ emissions and fine-scale winds responsible for plume dispersion. Furthermore, we cannot expect the satellite to match the exact spatial heterogeneity observed by the point measurements from Pandora because these instruments observe a very narrow fraction of the atmosphere and measure column $NO_2$ in a fundamentally different manner.

### 3.4.2 Comparison to the Airborne Compact Atmospheric Mapper (ACAM) spectrometer NO₂

The ACAM $NO_2$ instrument acquired measurements of tropospheric column $NO_2$ below altitudes of 8 km during DISCOVER-AQ Maryland. We match ACAM $NO_2$ measurements within ± one hour of the OMI overpass time to valid OMI $NO_2$ measurements. The comparison is plotted in Figure 8. Both the slope and $r^2$-value of the new OMI_CMAQ_OD product are closer to one when compared to the OMI_GMI product indicating that the OMI_CMAQ_OD product yields better agreement with ACAM $NO_2$. The low $r^2$-values may be related to the ACAM instrument random error, one of which is the use of unpolluted background spectra instead of reference spectra to process the data (Lamsal et al., 2017). In Figure 8a, we shade the points based on date. There were only six days in which valid OMI $NO_2$ spatially and temporally overlapped with the ACAM $NO_2$ data. In Figure 8b, we shade based on percentage coverage. Since the ACAM field of view is very small compared to OMI, pixel coverage from the ACAM would often only overlap a very small subset of the OMI pixel (median: 12 % of the OMI pixel). Since the ACAM is only measuring the portion of the tropospheric column below 8 km, there should be a consistent high bias in the OMI $NO_2$; instead there is a consistent low bias. This may be due to an artifact of the flight path of the UC-12 which preferentially sampled over the densest urban locations: OMI pixels are much larger in size and are capturing a more regional, and thus lower, value.

### 3.4.3 Comparison to the EPA NO₂ ground monitor network

The long-term EPA monitoring network provides surface observations outside the July 2011 timeframe. In Figure 9a, we compare mean $NO_2$* at each monitoring site to the two satellite products. All valid $NO_2$* data at each monitoring site over a 5-year (2008-2012) 2-month (June & July) period are averaged into a single point (up to 305 data entries) and matched to an average of satellite data over the same time period. The correlation between OMI_GMI and surface observations is $r^2 = 0.39$, while the correlation between OMI_CMAQ_OD and surface observations is $r^2 = 0.60$, a substantial improvement. This suggests that a high-resolution satellite product with improved AMFs, can detect surface $NO_2$ concentrations with more accuracy. As shown in Table 2, OMI_CMAQ without observational constraints performs almost as well ($r^2 = 0.55$); this is especially encouraging since comprehensive field measurements, such as those from DISCOVER-AQ, are limited in spatial and temporal scope.

### 3.5 OMI_CMAQ vs. CMAQ

We can now more fairly assess the $NO_2$ columns simulated by CMAQ using a high-resolution OMI $NO_2$ product. In Figure 10, we show a comparison between CMAQ and OMI_CMAQ_OD. Only model data within +/- 1 hour of and co-located with valid overpass data are shown in order to remove biases during cloudy days or days with invalid

data. We see a consistent model low bias in rural areas, and consistent model high bias in urban areas. Interestingly the high bias is larger in the immediate Baltimore metropolitan area compared to the D.C. metropolitan region.

We attribute the model low bias in rural regions to several shortcomings of this model simulation. This simulation did not include $NO_x$ emissions from soils. Rasool et al. (2016) has shown soil $NO_x$ emissions to be particularly important in the central United States, with a lesser role in the eastern United States. Excluding these emissions may have resulted in less $NO_x$ being transported from upwind regions. This model simulation utilized CB05 gas-phase chemistry, which is known to underestimate the recycling of alkyl nitrates back to $NO_2$ (Hildebrandt-Ruiz and Yarwood, 2013; Canty et al., 2015). CB05e51 gas-phase chemistry, released in a newer version of CMAQ (https://www.airqualitymodeling.org/in-dex.php/CMAQ_v5.1_CB05_updates), better handles alkyl nitrates and employs faster recycling of short-lived alkyl nitrate species. Faster recycling of alkyl nitrates would yield higher $NO_2$ concentrations throughout the modeling domain. Travis et al. (2016) found that upper tropospheric $NO_x$ is too low when compared to observations from aircraft during SEAC[4]RS. This is possibly due to downward stratospheric transport, outflow from convection, or OH chemistry that is not characterized correctly by models. Lightning $NO_x$ is still a very active area of research (Pickering et al., 2016). Although this model simulation did include lightning $NO_x$ emissions, there is a possibility these emissions are underestimated.

We attribute the model high bias in urban regions within our domain to an overestimate of anthropogenic $NO_x$ emissions (Anderson et al., 2014; Souri et al., 2016). This may be due to an improper allocation of area and mobile (on-road and off-road) source emissions which are spatially distributed based on population and number of cars respectively, or quite simply an overestimate of these sector emissions. Quantifying the uncertainty in MOVES, the mobile emissions software, is an active area of research.

**3.6 Comparison of model to satellite and in situ observations**

To further evaluate the model, we compare the model simulation to DISCOVER-AQ and EPA observations. In Figure 7b, we show a Pandora $NO_2$ comparison in the same manner as Figure 7a. In addition to showing CMAQ, we also show OMI_CMAQ_OD. We add the stratospheric VCD information from the OMI $NO_2$ Level 2 product to the CMAQ tropospheric columns to ensure a fair comparison. Both the model and new OMI $NO_2$ product have a slope close to unity indicating that both are able to match the variability in $NO_2$ columns. There is, however, a consistent low offset. This may indicate that the stratospheric VCD in the NASA Level 2 retrieval may be too low during this two-month timeframe. The $r^2$ of CMAQ is higher than OMI_CMAQ_OD. This is not particularly surprising because the resolution of the satellite is coarse, despite it being processed with new air mass factors.

In Figure 9b, we show a comparison between CMAQ, OMI_CMAQ_OD and ground monitors for June & July 2011. The $r^2$ between CMAQ and ground monitors is 0.70, while the correlation with the new satellite product is 0.73. The OMI_CMAQ_OD product has a better correlation with ground $NO_2$ monitors than the 1.33 km CMAQ simulation alone indicating that there is added utility in the satellite data.

**4 Summary and conclusions**

This study demonstrates the critical importance of using high-resolution a priori $NO_2$ shape factors to develop AMFs in and around metropolitan areas. We develop three new OMI $NO_2$ products: using high spatial resolution $NO_2$ profiles from a 1.33 km CMAQ model simulation (OMI_CMAQ), using CMAQ profiles constrained by in-situ observations (OMI_CMAQ_O), and applying model-derived spatial information (downscaling) to OMI_CMAQ_O (OMI_CMAQ_OD). When using high spatial resolution models to develop the AMF, the mean AMF in urban areas decreases by up to 50 % causing the tropospheric VCDs in urban areas to increase by up to a factor of two. This is because high-resolution models simulate larger concentrations near the surface in urban areas. In essence, we are reprocessing the satellite to look for $NO_2$ closer to the surface than in the original product. We believe this finding extends to other urban areas since coarse global models will consistently merge rural and urban pollution, and subsequently overestimate the AMF in city centers.

Another novel step in our re-processing technique is using in situ observations to enhance modeled $NO_2$ profile shapes. CMAQ $NO_2$ values in the Baltimore-Washington metropolitan region are generally too large within the boundary layer, too small in the mid-troposphere, and a factor of three too small in the uppermost troposphere. These particular biases may not be fully applicable to rural regions, since the DISCOVER-AQ field campaign was only focused in the urban corridor. As a result, our adjusted satellite product in rural regions may have higher uncertainty than urban areas. With that said, constraining model simulations to observations yields an improved satellite product over the non-constrained product when comparing to Pandora $NO_2$. Furthermore, by constraining to observations, we reduce the dependence on a priori emission inventories (e.g., NEI) used in model simulations, which can have deficiencies (Anderson et al., 2014; Souri et al., 2016; Travis et al., 2016). For example, in the constraint-based product, VCDs in Baltimore are 30 % lower than the OMI_CMAQ product. The tropospheric VCDs in rural Mid-Atlantic areas are 20 – 50 % lower than both the OMI_CMAQ and operational products. This is a particularly important finding because it means that the poor performance of CMAQ (or any model used for a satellite retrieval) will manifest itself in the retrieval. This will be a difficult challenge going forward, and emphasizes the need to use state-of-the science models for satellite retrievals.

Lastly, we apply a technique developed by Kim et al. (2016) to downscale OMI $NO_2$ data. This technique is especially valuable for pollutant exposure health studies, which require high-resolution long-term pollutant estimates. The downscaling procedure provides a higher spatial resolution snapshot of $NO_2$, while not altering the observed satellite pixel values. Instead, this technique re-allocates values across the pixel based on the variability within the high-resolution model. As a result, the new satellite product (OMI_CMAQ_OD) shows higher values in urban, polluted areas and lower values in rural, unpolluted areas than the operational OMI_GMI product. This new product better captures the urban-scale variability of $NO_2$ and has a much better correlation with ground monitors. A deficiency with this technique is that if a localized source, such as a power plant plume or wildfire, is not simulated at all by the model, then this error will be passed on to the product. Furthermore, if the area is affected by a mesoscale meteorological feature that is simulated incorrectly by the model, such as a thunderstorm, valley breeze,

or sea breeze, the model will be similarly deficient. Therefore, we do not recommend using the downscaling technique in areas where the emission inventory or meteorology is very uncertain.

We must clarify, however, that the results in this paper are only applicable to our region of interest. While we find that rural areas within our mid-Atlantic model domain now have tropospheric $NO_2$ columns which are $20 - 50$ %

lower than the operational product, we cannot conclude that this would be the same elsewhere. The "rural" locations within our model domain are situated in a particularly tricky spot because they are close, but not too close to major urban areas: a GMI simulation with a resolution of $1.25° \times 1°$ ($\sim 110 \times 110$ km) will group rural areas into a grid cell also including a large city. Therefore a location that is 100's of kilometers from the nearest city and with spatial homogeneity may be simulated with consistency by GMI and therefore the operational OMI product may be

an accurate representation of reality in these cases.

The refined OMI_CMAQ_OD product provides a better $NO_2$ column measurement when compared to Pandora column $NO_2$: the slope is near unity and the $r^2$ increases over the operational OMI $NO_2$ product. An important finding of this work is that using a high-resolution model, not the constraining to observations, provides the majority of the improvement, when comparing to ground monitors. This suggests that a high-resolution model with

reasonable fidelity can be used anywhere in the world, and is not tied into an area in which a field experiment is located.

This technique can be used as a bridge until newer instruments such as TROPOMI are instituted. Future instruments will have increased spatial resolution, but comparison to OMI without using this technique may yield large differences around urban areas. At the same time, we demonstrate the importance of using high-resolution and high-

fidelity model simulations for retrievals from future satellite missions. A combination of both increased satellite resolution and model resolution are needed in order to improve $NO_2$ satellite retrievals. We urge other community members to generate high-resolution OMI $NO_2$ data using this technique if it is to be used for small-scale ($< 100$ km length scale) studies as it provides a better alternative for urban areas than standard satellite products.

**Acknowledgments**

This publication was developed under Assistance Agreement No. RD835871 awarded by the U.S. Environmental Protection Agency to Yale University. It has not been formally reviewed by EPA. The views expressed in this document are solely those of the authors and do not necessarily reflect those of the Agency. EPA does not endorse any products or commercial services mentioned in this publication. We would like to thank two anonymous reviewers for their constructive comments in improving this work. We would like to thank Ron Cohen of UC-

Berkeley and his research group for their observations of $NO_2$ from the P3-B aircraft during DISCOVER-AQ Maryland. We would also like to thank Jay Herman of UMBC and NASA Goddard Space Flight Center and his research group for their Pandora $NO_2$ measurements during this same time period. All data from DISCOVER-AQ Maryland can be downloaded freely from http://www-air.larc.nasa.gov/cgi-bin/ArcView/discover-aq.dc-2011. EPA $NO_2$* data was downloaded from the AQS Data Mart, and can be freely retrieved from:

https://aqs.epa.gov/aqsweb/documents/data_mart_wel-come.html. We would also like to thank Chinmay Satam formerly at the University of Maryland and now at Georgia Tech for preparation of the emissions in the CMAQ model simulation. We acknowledge the free use of $NO_2$ column data from the OMI sensor available at: https://disc.gsfc.nasa.gov/Aura/data-holdings/OMI/omno2_v003.shtml. Argonne National Laboratory is operated by UChicago Argonne, LLC, under contract no. DE-AC2-06CH11357 with the U.S. Department of Energy.

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

**Figures**

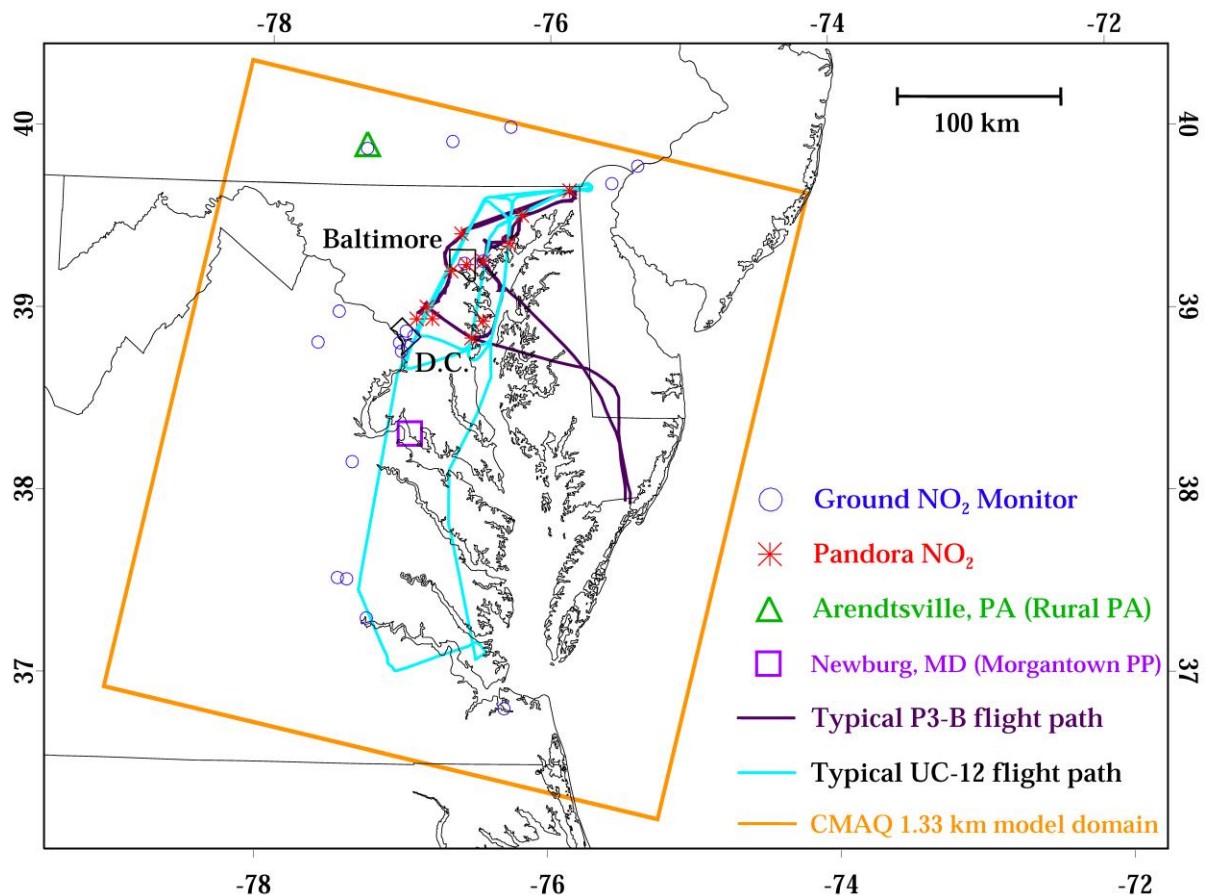

**Figure 1.** The Mid-Atlantic United States: the area of interest for this research project. Model domain and observation locations are depicted. There are eighteen EPA chemiluminescence $NO_2$ monitors and twelve Pandora $NO_2$ measurement sites.

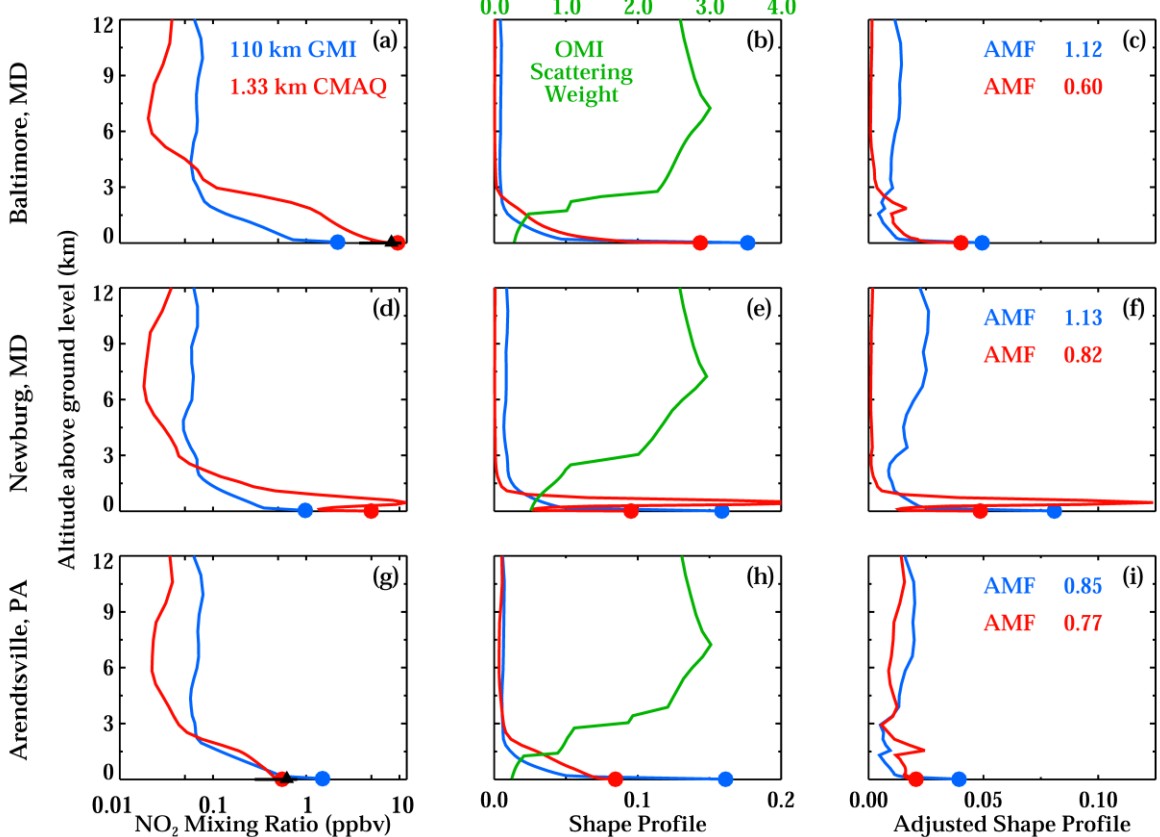

**Figure 2**. (**a, d, g**) Mean 2 PM local time June and July 2011 NO$_2$ mixing ratio as a function of altitude from a GMI (1.25° × 1°; ~110 ×110 km) model simulation and CMAQ (1.33 × 1.33 km) model simulation for three locations: (**a**) downtown Baltimore, (**d**) Morgantown power plant in Newburg, MD and (**g**) Arendtsville in rural Pennsylvania. Black triangles with error bars as discussed in the text represent co-located surface observations from the EPA monitoring network. (**b, e, h**) NO$_2$ shape profiles (partial NO$_2$ columns divided by total NO$_2$ column) as function of altitude for the same timeframe and locations; green line denotes co-located OMI scattering weight. (**c, f, i**) "Adjusted" shape profiles, partial NO$_2$ columns divided by total NO$_2$ columns multiplied by OMI scattering weight, as function of altitude for the same timeframe and locations.

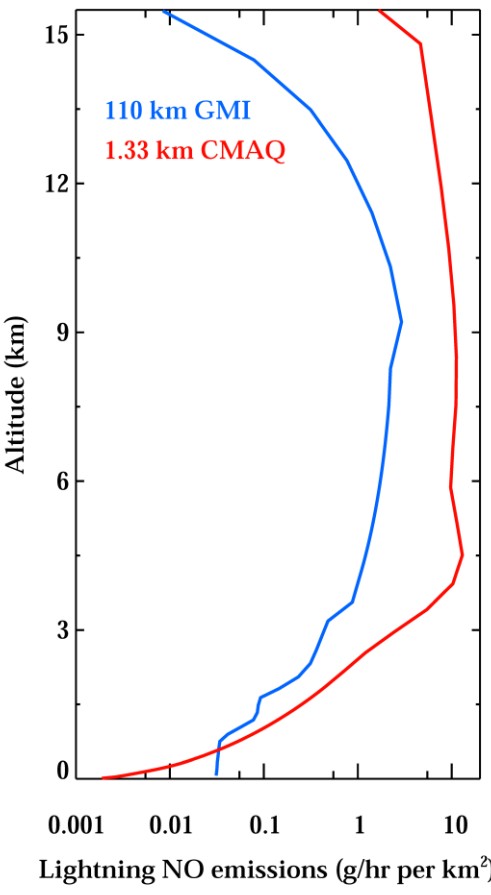

**Figure 3**. Mean June and July 2011 lightning NO emissions as a function of altitude from the GMI (1.25° × 1°; ~110 ×110 km) and CMAQ (1.33 × 1.33 km) model simulations.

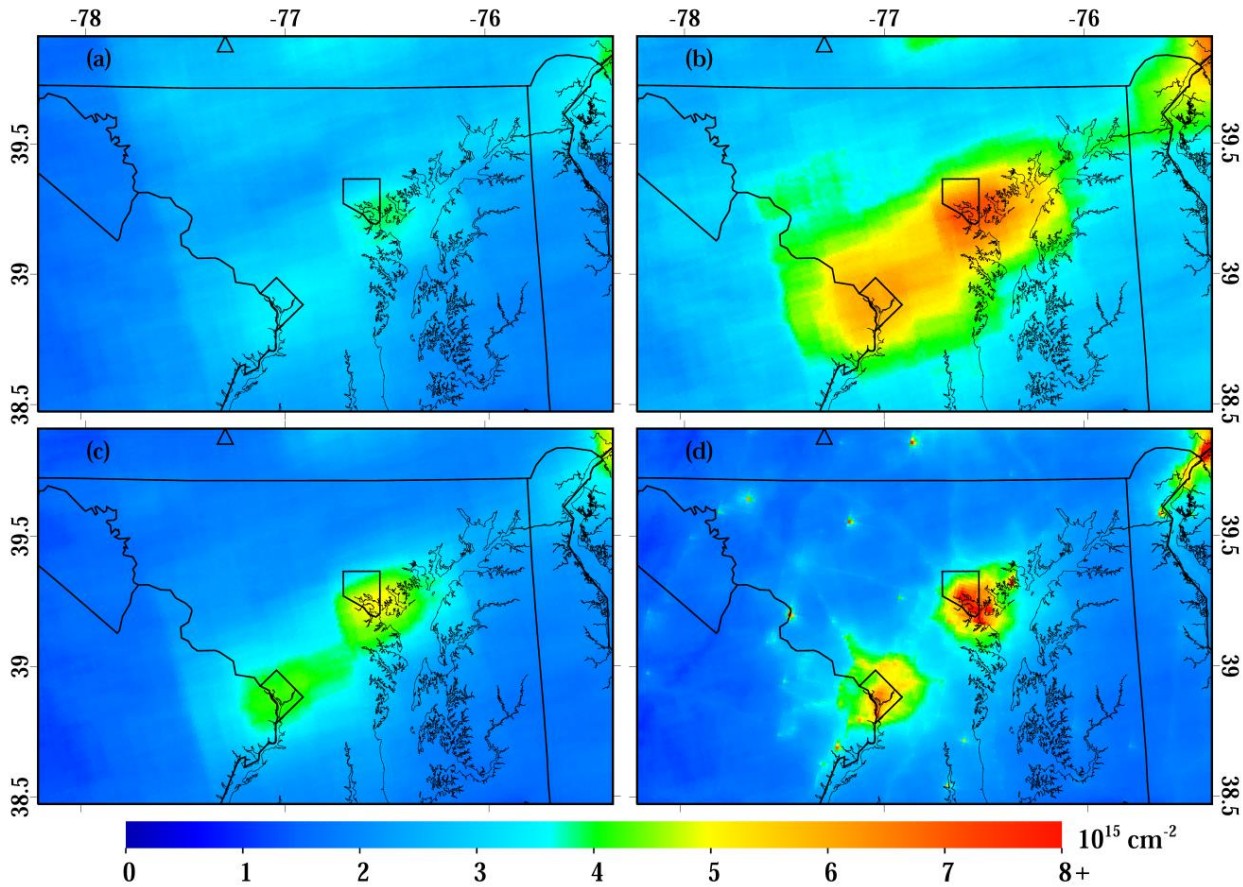

**Figure 4.** Oversampled OMI NO$_2$ tropospheric columns at 1.33 km resolution in the Baltimore-Washington metropolitan area for June & July 2008 – 2012 (2 months × 5 years; 10 months total). **(a)** The NASA version 3.0 operational OMI NO$_2$ product using GMI NO$_2$ shape profiles (OMI_GMI). **(b)** OMI NO$_2$ using CMAQ a priori NO$_2$ shape profiles (OMI_CMAQ). **(c)** OMI NO$_2$ using CMAQ a priori NO$_2$ shape profiles constrained by observations (OMI_CMAQ _O). **(d)** OMI NO$_2$ using CMAQ a priori NO$_2$ shape profiles constrained by observations and spatial weighting downscaling kernels (OMI_CMAQ_OD). In all plots, Arendtsville, PA is denoted by the triangle in the top left corner.

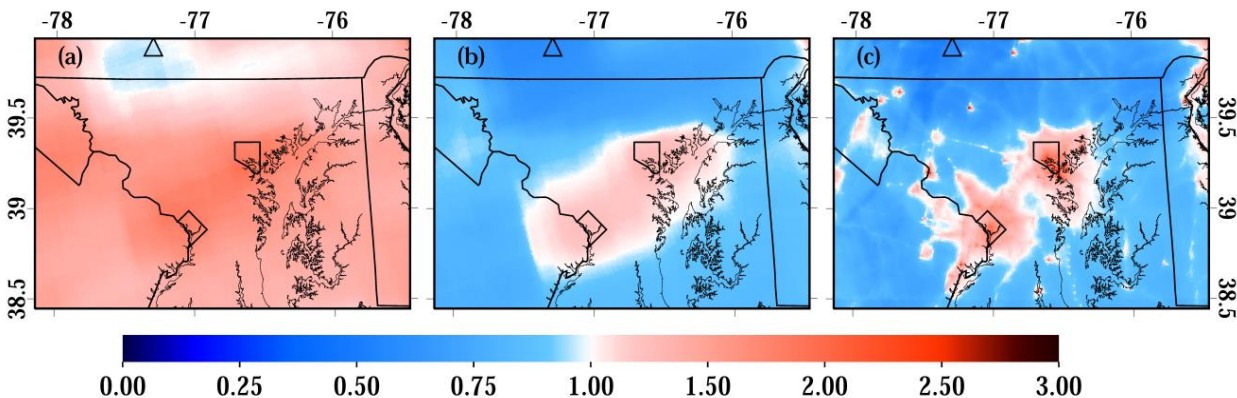

**Figure 5.** Ratio between the three OMI_CMAQ tropospheric NO$_2$ retrievals and the operational NASA v3.0 OMI tropospheric NO$_2$ retrieval for June & July 2008 – 2012 (2 months × 5 years; 10 months total). **(a)** OMI_CMAQ / OMI_GMI. **(b)** OMI_CMAQ_O/ OMI_GMI. **(c)** OMI_CMAQ_OD/ OMI_GMI.

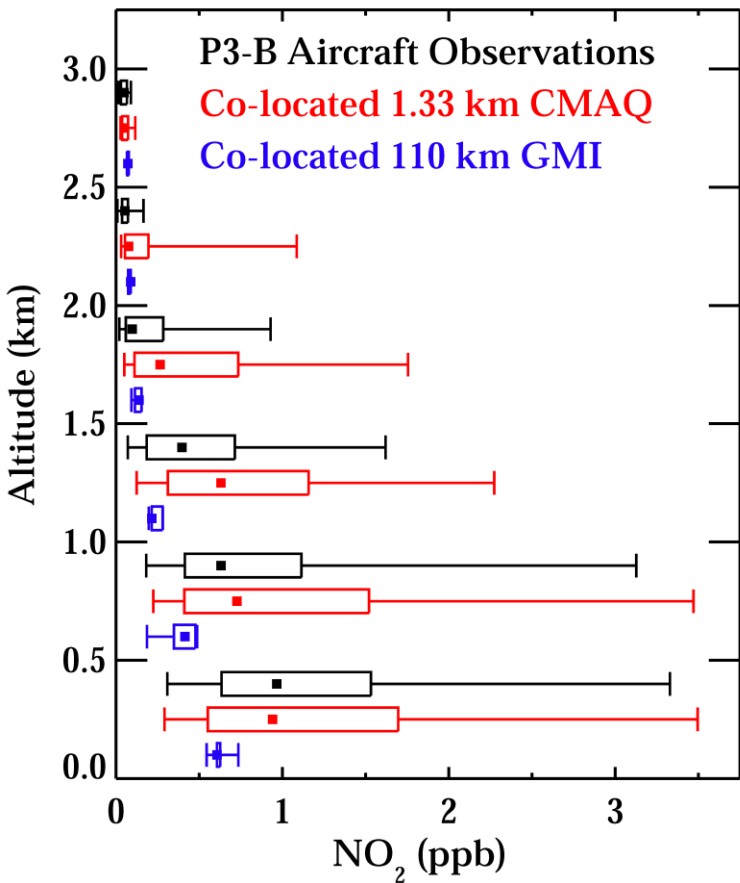

**Figure 6.** Vertical profiles of NO$_2$ binned in 500 m intervals (0 – 0.5 km, 0.5 – 1km, etc.) showing the 5$^{th}$, 25$^{th}$, 50$^{th}$, 75$^{th}$, and 95$^{th}$ percentiles within ± 2 hours of the OMI overpass time. **(Black)** One minute averaged data from the P3-B aircraft during July 2011 DISCOVER-AQ Maryland. **(Red)** Model output from CMAQ matched spatially and temporally to the P3-B measurements at 1 min intervals. **(Blue)** July 2011 monthly mean model output from GMI matched spatially to the P3-B measurements at 1 min intervals. In all cases, the squares indicate the median values.

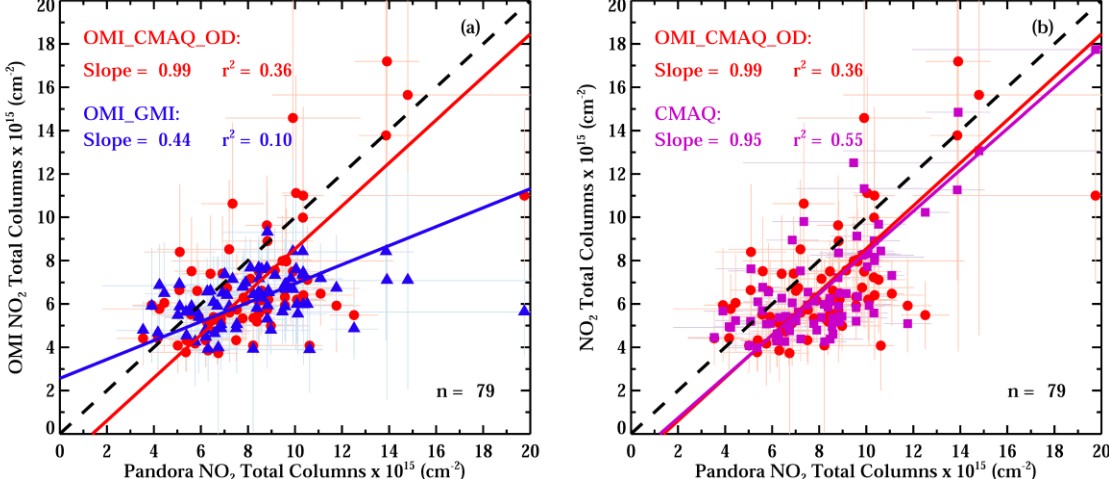

**Figure 7.** (a) Total column NO$_2$ OMI_GMI and OMI_CMAQ_OD versus co-located spatially and temporally Pandora NO$_2$ total column measurements within ± 1 hour of a valid satellite overpass during July 2011. (b) Same but now showing CMAQ instead of OMI_GMI; the stratospheric vertical column from NASA Level 2 product has been added to CMAQ to ensure a fair comparison. Error bars on both plots represent ± one standard deviation away from the mean**.**

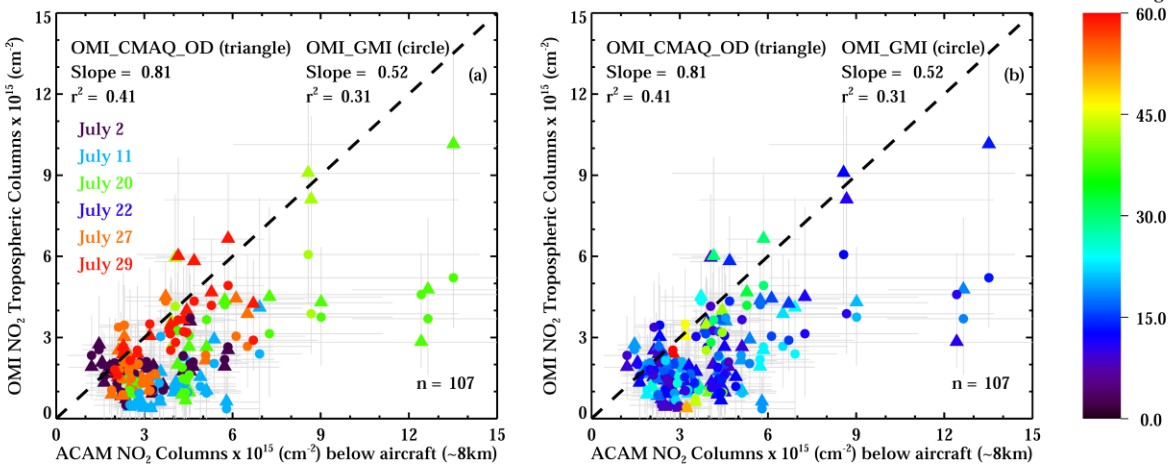

**Figure 8.** Tropospheric column NO$_2$ OMI_GMI and OMI_CMAQ_OD versus co-located spatially and temporally matched ACAM NO$_2$ column measurements within ± 1 hour of a valid satellite overpass during July 2011. **(a)** Color-coded based on date. **(b)** Color-coded based on percent coverage. Error bars on both plots represent ± one standard deviation away from the mean**.**

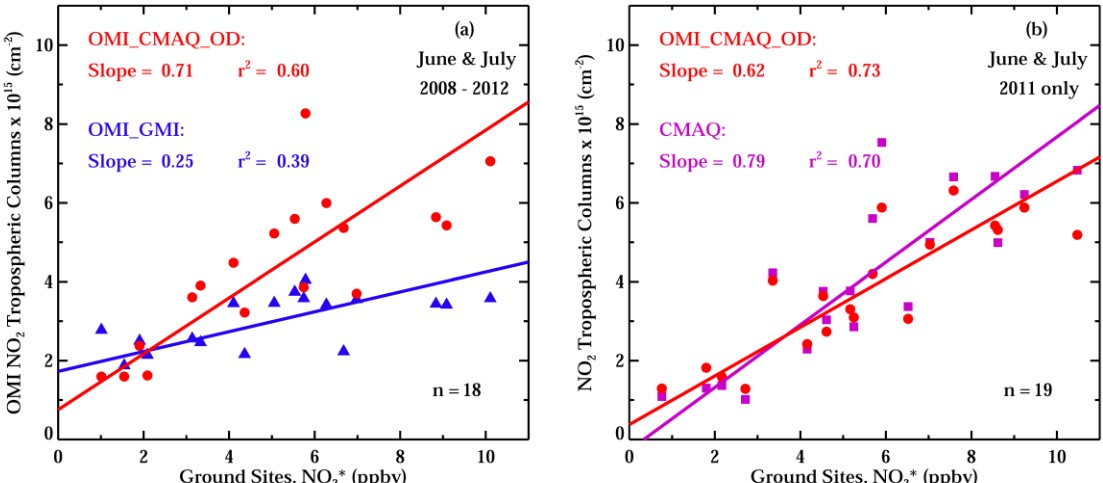

**Figure 9.** **(a)** Tropospheric column $NO_2$ OMI_GMI and OMI_CMAQ_OD versus co-located ground $NO_2$* chemiluminescence measurements within ± 2 hours of a valid satellite overpass during June & July 2008 through 2012; all ~300 daily ground monitor values are merged into a mean single value and compared to the satellite mean over the same corresponding time period. **(b)** Same as (a) but now comparing CMAQ and OMI_CMAQ_OD for June & July 2011 only.

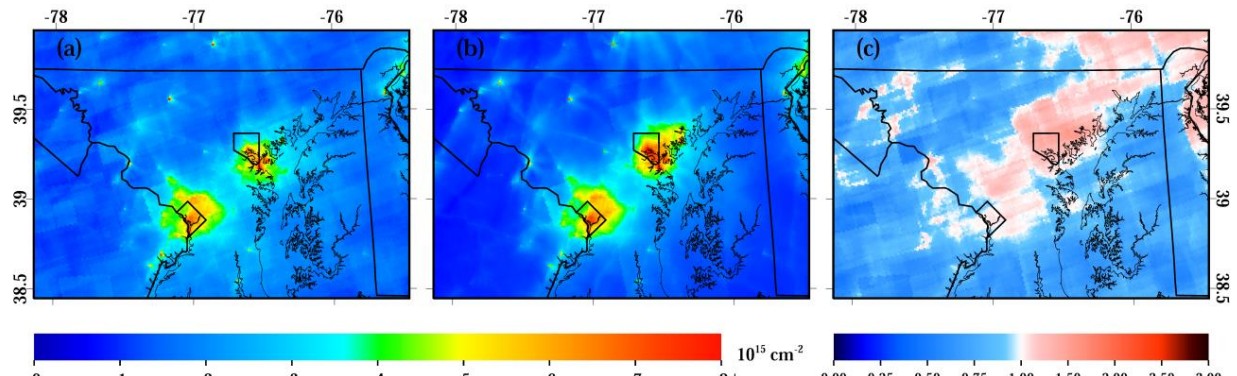

**Figure 10.** Oversampled tropospheric column $NO_2$ at 1.33 km in the Baltimore-Washington metropolitan area for June & July 2011 only. **(a)** OMI_CMAQ_OD. **(b)** CMAQ $NO_2$ corresponding to valid overpass times. **(c)** Ratio between the two plots CMAQ / OMI_CMAQ_OD.

**Table 1.** Summary of the current OMI NO$_2$ retrievals in the literature.

| | NASA OMNO2 v3 | DOMINO v2 | BeHR NO$_2$ | POMINO | HKOMI NO$_2$ | This study |
|---|---|---|---|---|---|---|
| **CTM** | GMI Global 1° × 1.25° | TM4 Global 2° × 3° | WRF-Chem U.S. 12 × 12 km | GEOS-Chem China 0.667° × 0.5° | WRF-CMAQ PRD China 3 × 3 km | WRF-CMAQ East U.S. 1.33 × 1.33 km |
| **RTM** | TOMRAD | DAK | TOMRAD | LIDORT v3.6 | SCIATRAN | TOMRAD |
| **A priori NO$_2$ profile** | Monthly mean profiles | Monthly mean profiles | Daily profiles when it exists. Monthly mean profiles elsewhere. | Monthly mean profiles | Daily profiles | Daily profiles when it exists. Monthly mean profiles elsewhere. All profiles constrained to aircraft observations. |
| **Surface pressure** | MERRA downscaled to 90 arcsec DEM | TM4 downscaled to 3 × 3 km | WRF downscaled to 1 × 1 km using GLOBE | GEOS-5 0.667° × 0.5° | WRF 3 × 3 km | WRF 1.33 × 1.33 km |
| **Surface albedo** | OMI LER climatology 0.5° × 0.5° taken from Kleipool et al., 2008 | OMI LER climatology 0.5° × 0.5° taken from Kleipool et al., 2008 | MODIS black-sky albedo MCD43C2 at 0.05° × 0.05° | Over land: MODIS BRDF MCD43C2 at 0.05° × 0.05° Over ocean: OMI LER taken from Kleipool et al., 2008 | MODIS MCD43C2 at 0.01° × 0.01° | OMI LER climatology 0.5° × 0.5° taken from Kleipool et al., 2008 |
| **Aerosol correction** | Implicitly corrected through cloud products | Implicitly corrected through cloud products | Implicitly corrected through cloud products | Explicit treatment of aerosols | Correction for the aerosol effect | Implicitly corrected through cloud products |

**Table 2.** Slope and r$^2$ for all four OMI satellite products compared to Pandora NO$_2$ from July 2011 and EPA ground monitor NO$_2$* observations from June & July 2008 – 2012. Pandora NO$_2$ is compared to the OMI NO$_2$ total column products, while the EPA ground monitors are compared to OMI NO$_2$ tropospheric column products. Figures 7a and 9a show values for OMI_GMI and OMI_CMAQ_OD only.

| | Pandora NO$_2$ | | EPA NO$_2$* | |
|---|---|---|---|---|
| | Slope | r² | Slope | r² |
| OMI_GMI | 0.44 | 0.10 | 0.25 | 0.39 |
| OMI_CMAQ | 1.23 | 0.12 | 0.54 | 0.55 |
| OMI_CMAQ_O | 0.64 | 0.18 | 0.41 | 0.57 |
| OMI_CMAQ_OD | 0.99 | 0.36 | 0.71 | 0.60 |