# Peer review of "A HIGH-RESOLUTION AND OBSERVATIONALLY CONSTRAINED OMI NO2 SATELLITE RETRIEVAL"

_Atmospheric Chemistry and Physics, 2017_

## Referee Comment (RC1) · Anonymous Referee #1 · 24 Apr 2017

Review of "A High-Resolution and Observationally Constrained OMI NO2 Retrieval", ACPD, by Daniel L. Goldberg, Lok N. Lamsal, Christopher P. Loughner, Zifeng Lu and David G. Streets

The current paper presents a modified product of tropospheric NO2 columns during the Maryland 2011 DISCOVER-AQ. One of the major uncertainties of retrieving the vertical NO2 columns originate from the calculation of AMFs. The uncertainty of AMF might resulted from several inputs, but here, the main focus is on the shape factors. The authors claimed that a high resolution model output can potentially distinguish highly polluted regions from others, because they are roughly mixed in the original OMI product. The authors further brought up an important issue in CMAQ that is its large underprediction of NO2 in the free troposphere. This issue has nicely been addressed by using P3-B measurements to constrain the CMAQ NO2 profiles. Finally, the authors

made use of CMAQ to downscale OMI tropospheric NO2 columns to provide a very high resolution "map" using the method of Kim et al., [2016]. Although there is not a great deal of effort to advance the retrieval process, I believe that this will be interesting for environmental agencies who are looking for a very fine product, particularly for the use of health impacts. This manuscript in its present form requires significant improvement before being acceptable for publication in ACP.

The first major problem with this study is the lack of adequate comparison in terms of magnitude. I am aware that the authors used the Pandora measurements, but the comparison might have been influenced by errors in OMI stratospheric NO2 columns (which had been added to the tropospheric ones to conduct an apples-to-apples comparison with Pandora). Since you are using a more spatially detailed model to estimate shape factors, the improvement in correlation is expected. However, it is imperative to use other products such as ACAM [Lamsal et al., 2017, JGR] to show if the new product will get closer to observations with small footprints. My suggestion will become more serious for your last product (OMI_CMAQ_OD). This is a pseudo observation based on integrating a model and an observation. Thus, its accuracy should be more carefully verified.

My second major concern is about overlooking the impact of accurate NO2 (and other gases) during estimating the scattering weights. Two parts that the priori NO2 values from model are used during the retrieval are i) estimating the jacobian values from a radiative transfer model and ii) calculating the shape factors. Why the first step was not performed or not even mentioned in this paper? Would considering a more accurate a NO2 priori profile lead to a better estimation of scattering weights? This may be investigated using VLIDORT.

My specific comments follow:

P2, Line 18. "to generate tropospheric air mass factors..." might be changed to "to re-calculate" or "to modify" OMI tropospheric air mass factors.

P2, Line 24. How about the bias compared to ACAM or other air-borne observations?

P3, Line 13. "RO2" instead of "HO2" would be broader. You might need to define it in parentheses.

P4, Line10. I suggest the authors mention about the recent changes in China in 2011-2012 [Souri et al., 2017] mostly due to using SCR for power plants. Souri, A.H., Choi, Y., Jeon, W., Woo, J.H., Zhang, Q. and Kurokawa, J.I., 2017. Remote sensing evidence of decadal changes in major tropospheric ozone precursors over East Asia. Journal of Geophysical Research: Atmospheres, 122(4), pp.2474-2492.

P4, Line 14. How about the nadir-spectrometers like TES? This sentence might be revised. The footprint of surface concentrations exists in OMI signal. But it is not easy to separate it. The current sentence leaves readers with an impression that the radiance has not been impacted by the surface concentrations at all.

P4, Line35. You might want to elaborate their works in the introduction.

P5, Line 22. How was the stratospheric slant column subtracted from total column in OMI? CTMs or based on the OMI radiance?

P5, Line 26. I would suggest adding Martin et al., 2002 for NO2 shape profile.

P5, Line 30. Please provide references. I am assuming that scattering weights are already stored in a six-dimension LUT, and for partially cloudy pixels, a lambertian surface with albedo equal to 0.8 is assumed, then they combine the results (cloudy and clear) using the IPA. Is the new product different from this?

P6, Line 1. Please clarify whether the profile from GMI model is constant over time. You may need to mention: "It should be noted that a blue light converted which selectively photolyzes NO2 was used for P3-B. As a result, there was no need to modify NO2 concentration by applying an empirical equation from [Lamsal et al., 2008]."

P7, Line 4. NO2 varies quickly by time, and using a short duration is more appropriate,

because OMI captures NO2 just in a matter of milliseconds. Please check whether reducing the time average will make the comparisons better.

P8, Line3. Please specifically mention which scheme was used for the biogenic emissions. Did the authors consider the soil NOx emissions?

Figure 2. Why no observations were shown? In the text, you claimed that CMAQ has a better performance compared to observations (P9, line1).

P9, Line 10. I am not sure both model used the same lighting NOx option. The way they treat lightning might differ. There exit myriad of reasons for the underprediction of CMAQ NO2. It can be related to vertical mixing, uncertainty from NOx aviation emissions, stratospheric sources, or lightning. The vertical allocation of emissions are also different. If you had used the GMI for the chemical boundary conditions of CMAQ, it would have been easier to compare them.

P9, Line 19. Poor grammar.

P9, Line23. This is a very important message. It means the poor performance of CMAQ in simulating NO2 in free troposphere will make a challenge for the retrieval purposes. We may have to use the aircraft to constrain it, or to use GMI models at those specific altitudes. You may want to highlight it in the conclusion.

P11, Line 18. Please explain why it is rudimentary (i.e., not considering the errors in observations, model and the priori).

P11, Line 30. The discussion is not enough. Please explain the possible reasons of large differences between OMI_CMAQ and OMI_CMAQ_O. Were AMFs enhanced largely due to larger shape factors in the free troposphere?

P12, Line 28. I don't agree with your sentence "OMI can now "see" ...". This is an illusion. You used the model to downscale the values. This is not the OMI; it is the model that provided a tool to concentrate the observations. I would call it a pseudo observation or simply a "map". We have to clarify that OMI footprint is too coarse to

see these plumes. That's why we need TEMPO and TROPOMI. This paragraph should be revised or be dropped.

P15, Line 30. It is not only about the emissions, but also the meteorological fields. Simulating surface winds in many situations is not straightforward. So the winds may be off in the model resulting in wrong distribution of final product.

---

## Author Comment (AC1)

To both reviewers:

Thank you for your constructive comments. They have helped to improve this manuscript. Here is a short description of the changes to the Figures and Tables. More in-depth comments are in the responses to each reviewer.

*Figure 1 has been modified to include the UC-12 aircraft flight path.*

Figure 3 has been added based on Reviewer #2's suggestion.

Figure 6 (previously Figure 5) has been modified to include NO2 vertical profiles from GMI.

Figure 7 (previously Figure 6) has been modified to include only Pandora measurements. Figure 8a in the original manuscript is now Figure 7b. We also made some cosmetic changes, which include adding the standard deviation.

Figure 8 has been added based on Reviewer #1's suggestion.

Figure 9 (previously Figure 8) has been modified to include only EPA measurements. Figure 6b in the original manuscript is now Figure 9a. We also made some cosmetic changes.

Table 1 has been added. It compares the OMI NO2 retrieval developed during this study to other OMI NO2 retrievals.

**Anonymous Referee #1**

Review of "A High-Resolution and Observationally Constrained OMI NO2 Satellite Retrieval", ACPD, by Daniel L. Goldberg, Lok N. Lamsal, Christopher P. Loughner, Zifeng Lu and David G. Streets

The current paper presents a modified product of tropospheric NO2 columns during the Maryland 2011 DISCOVER-AQ. One of the major uncertainties of retrieving the vertical NO2 columns originate from the calculation of AMFs. The uncertainty of AMF might resulted from several inputs, but here, the main focus is on the shape factors. The authors claimed that a high resolution model output can potentially distinguish highly polluted regions from others, because they are roughly mixed in the original OMI product. The authors further brought up an important issue in CMAQ that is its large underprediction of NO2 in the free troposphere. This issue has nicely been addressed by using P3-B measurements to constrain the CMAQ NO2 profiles. Finally, the authors made use of CMAQ to downscale OMI tropospheric NO2 columns to provide a very high resolution "map" using the method of Kim et al., [2016]. Although there is not a great deal of effort to advance the retrieval process, I believe that this will be interesting for environmental agencies who are looking for a very fine product, particularly for the use of health impacts. This manuscript in its present form requires significant improvement before being acceptable for publication in ACP.

\*The first major problem with this study is the lack of adequate comparison in terms of magnitude. I am aware that the authors used the Pandora measurements, but the comparison might have been influenced by errors in OMI stratospheric NO2 columns (which had been added to the tropospheric ones to conduct an apples-to-apples comparison with Pandora). Since you are using a more spatially detailed model to estimate shape factors, the improvement in correlation is expected. However, it is imperative to use other products such as ACAM [Lamsal et al., 2017, JGR] to show if the new product will get closer to observations with small footprints. My suggestion will become more serious for your last product (OMI\_CMAQ\_OD). This is a pseudo observation based on integrating a model and an observation. Thus, its accuracy should be more carefully verified.

We conducted a comparison with the ACAM data. The ACAM NO2 product is introduced in section 2.2.3 and the analysis is described in section 3.4.2. Figure 8 describes the requested comparison: ACAM NO2 vs. the OMI\_GMI and our new OMI\_CMAQ\_OD NO2 product. The flight path of the UC-12 aircraft carrying the ACAM has been added to Figure 1.

\*My second major concern is about overlooking the impact of accurate NO2 (and other gases) during estimating the scattering weights. Two parts that the priori NO2 values from model are used during the retrieval are i) estimating the jacobian values from a radiative transfer model and ii) calculating the shape factors. Why the first step was not performed or not even mentioned in this paper? Would considering a more accurate a NO2 priori profile lead to a better estimation of scattering weights? This may be investigated using VLIDORT.

For this study, we follow previous studies and assume that scattering weights are a function of a single a priori  $NO_2$  profile (e.g., Palmer et al., 2001, Martin et al., 2002, Boersma et al., 2011, Bucsela et al., 2013). Therefore, we assume scattering weights and  $NO_2$  shape factors are independent. We are aware that at very high  $NO_2$  concentrations this assumption may not be fully valid. However, this is a novel and emerging research topic that is beyond the scope of this paper.

We now clarify this in Section 2.1 to say:

"The optical atmospheric/surface properties are characterized by the scattering weight (SW) and are calculated by a forward radiative transfer model (TOMRAD) which are output as a look-up table (LUT). The SWs are then adjusted real-time depending on observed viewing angle, surface albedo, cloud fraction, and cloud height. For this study, we follow previous studies (e.g., Palmer et al., 2001, Martin et al., 2002, Boersma et al., 2011, Bucsela et al., 2013) and assume that SWs and NO2 profile shapes are independent."

My specific comments follow:

\*P2, Line 18. "to generate tropospheric air mass factors..." might be changed to "to re-calculate" or "to modify" OMI tropospheric air mass factors.

**This has been changed as suggested.**

\*P2, Line 24. How about the bias compared to ACAM or other air-borne observations?

**This has been added to the abstract and addressed more fully in Section 3.4.2**

\*P3, Line 13. "RO2" instead of "HO2" would be broader. You might need to define it in parentheses.

**$RO_2$ is now included and defined. $HO_2$ is now also defined.**

\*P4, Line10. I suggest the authors mention about the recent changes in China in 2011- 2012 [Souri et al., 2017] mostly due to using SCR for power plants. Souri, A.H., Choi, Y., Jeon, W., Woo, J.H., Zhang, Q. and Kurokawa, J.I., 2017. Remote sensing evidence of decadal changes in major tropospheric ozone precursors over East Asia. Journal of Geophysical Research: Atmospheres, 122(4), pp.2474-2492.

**A short clarification on the increasing OMI NO2 trend in China before 2011, stabilization in 2011-2012, and decreases since 2012 has been added. The Souri et al. reference has also been added.**

We now state: "Over this 10-year period China has seen a reversal of its trends: during 2005-2010 OMI NO2 tropospheric columns were increasing (Verstraeten et al., 2015), in 2011-2012 they had stabilized (Souri et al., 2017), and since 2012 they have subsequently decreased as the country enforces its Twelfth 5-year plan (de Foy et al., 2016b)."

\*P4, Line 14. How about the nadir-spectrometers like TES? This sentence might be revised. The footprint of surface concentrations exists in OMI signal. But it is not easy to separate it. The current sentence leaves readers with an impression that the radiance has not been impacted by the surface concentrations at all.

**We have clarified this sentence to say:**

"Remote sensing instruments typically measure the entire column content instead of in situ concentrations at individual vertical levels. Being able to derive surface concentrations from column content information would be very useful for the policy-making and health-assessment communities."

\*P4, Line35. You might want to elaborate their works in the introduction.

We have added a full paragraph at the end of the Introduction explaining these three retrievals and their important findings. We have also now included this information in Table 1.

\*P5, Line 22. How was the stratospheric slant column subtracted from total column in OMI? CTMs or based on the OMI radiance?

The stratospheric slant column was subtracted based on OMI radiance (Bucsela et al., 2013). This has been clarified in the text:

"Stratospheric SCD... is inferred using a local analysis of the stratospheric field (Bucsela et al., 2013)"

\*P5, Line 26. I would suggest adding Martin et al., 2002 for NO2 shape profile.

**This reference has been added as suggested.**

\*P5, Line 30. Please provide references. I am assuming that scattering weights are already stored in a six-dimension LUT, and for partially cloudy pixels, a lambertian surface with albedo equal to 0.8 is assumed, then they combine the results (cloudy and clear) using the IPA. Is the new product different from this?

**Yes, the scattering weights are stored in a look-up table.**

This has been clarified in the text to say: "The optical atmospheric/surface properties are characterized by the scattering weight (SW) and are calculated by a forward radiative transfer model (TOMRAD), which are output as a look-up table (LUT). The SWs are then adjusted real-time depending on observed viewing angle, surface albedo, cloud fraction, and cloud pressure. For this study, we follow previous studies (e.g., Palmer et al., 2001, Martin et al., 2002, Boersma et al., 2011, Bucsela et al., 2013) and assume that SWs and NO2 profile shapes are independent."

\*P6, Line 1. Please clarify whether the profile from GMI model is constant over time. You may need to mention: "It should be noted that a blue light converted which selectively photolyzes NO2 was used for P3-B. As a result, there was no need to modify NO2 concentration by applying an empirical equation from [Lamsal et al., 2008]."

**This has been clarified to state a "monthly-averaged and year-specific" simulation was used.**

The second part of this comment appears to be in reference to P6, Line 27. In this section, we have added a clarification that the Cohen group instrument does not suffer from the same positive bias as chemiluminescence detectors, as suggested.

\*P7, Line 4. NO2 varies quickly by time, and using a short duration is more appropriate, because OMI captures NO2 just in a matter of milliseconds. Please check whether reducing the time average will make the comparisons better.

We have found that +/- 1 hour is the "sweet spot", in which there is minimal compromise in temporal matching, while also maintaining a large enough sample size. Typically, winds are 10 - 20 km/hr, so in essence, a pixel 20 - 40 km wide will be captured within a 2 hour window (assuming NO2 remains relatively constant over the 2-hour window). If we shorten the averaging time, then we severely limit the number of samples, which are already small. This means that if the Pandora instrument is sampling a local plume (or lack thereof) not representative of the nearby environment, then this will cause an unfair comparison.

\*P8, Line3. Please specifically mention which scheme was used for the biogenic emissions. Did the authors consider the soil NOx emissions?

**BEISv3.14 was used for the biogenic emissions. The soil NOx emissions parameterization was not released until after completion of this CMAQ simulation, so no soil NOx emissions are included in this CMAQ simulation. Both are clarified in the text.**

\*Figure 2. Why no observations were shown? In the text, you claimed that CMAQ has a better performance compared to observations (P9, line1).

Surface observations are shown on Figure 2 and are denoted by the black triangle. This is the original basis of the claim. However, we realized that this claim was very tenuous at best, so we have now compared CMAQ, GMI, and the P3-B aircraft observations on Figure 5. This figure demonstrably illustrates that CMAQ is better at simulating NO2 in urban areas than GMI.

\*P9, Line 10. I am not sure both model used the same lightning NOx option. The way they treat lightning might differ. There exit myriad of reasons for the underprediction of CMAQ NO2. It can be related to vertical mixing, uncertainty from NOx aviation emissions, stratospheric sources, or lightning. The vertical allocation of emissions are also different. If you had used the GMI for the chemical boundary conditions of CMAQ, it would have been easier to compare them.

**We have no re-phased the section to say:**

"To determine whether lightning NO is the primary driver of this difference, we compare lightning NO emissions from both model simulations in Figure 3. The CMAQ model ingests lightning NO emissions that are an order of magnitude larger than the GMI simulation at most altitudes. This is likely due to WRF simulating more convective precipitation and higher cloud-top heights, both input variables to the lightning NO parameterization, than GMI. Therefore, the smaller magnitude of free tropospheric NO2 in CMAQ does not arise from the lightning NOx parameterization, but instead from a combination of the chemistry, aviation emissions, vertical mixing, long-range transport, and stratospheric-tropospheric exchange."

\*P9, Line 19. Poor grammar.

**This has been corrected.**

\*P9, Line23. This is a very important message. It means the poor performance of CMAQ in simulating NO2 in free troposphere will make a challenge for the retrieval purposes. We may have to use the aircraft to constrain it, or to use GMI models at those specific altitudes. You may want to highlight it in the conclusion.

Yes, this is an extremely important point, and perhaps was not highlighted enough in the original manuscript. This is now mentioned in the abstract and is highlighted again in the conclusion. In the conclusion we state: "... the poor performance of CMAQ (or any model used for a satellite retrieval) will manifest itself in the retrieval. This will be a difficult challenge going forward, and emphasizes the need to use state-of-the science models for satellite retrievals."

\*P11, Line 18. Please explain why it is rudimentary (i.e., not considering the errors in observations, model and the priori).

**We have revised to say "simplified" instead of "rudimentary".**

\*P11, Line 30. The discussion is not enough. Please explain the possible reasons of large differences between OMI\_CMAQ and OMI\_CMAQ\_O. Were AMFs enhanced largely due to larger shape factors in the free troposphere?

**A short discussion has now been included at the end of Section 3.3.2.**

"The large reduction in NO2 tropospheric vertical columns between OMI\_CMAQ and OMI\_CMAQ\_O is an outcome of using larger AMFs. The larger AMFs are a result of the original overestimate within the boundary layer and underestimate in the free troposphere. This is a particularly important finding because it means that a model with large biases in the simulation of NO2 can yield poor tropospheric vertical column contents, despite high spatial resolution. This emphasizes the need to evaluate the emissions and chemistry of a model before it should be used for satellite retrievals."

\*P12, Line 28. I don't agree with your sentence "OMI can now "see" ...". This is an illusion. You used the model to downscale the values. This is not the OMI; it is the model that provided a tool to concentrate the observations. I would call it a pseudo observation or simply a "map". We have to clarify that OMI footprint is too coarse to see these plumes. That's why we need TEMPO and TROPOMI. This paragraph should be revised or be dropped.

**This has been revised to say the "spatially downscaled OMI product" instead of OMI**

\*P15, Line 30. It is not only about the emissions, but also the meteorological fields. Simulating surface winds in many situations is not straightforward. So the winds may be off in the model resulting in wrong distribution of final product.

This has been clarified in the text to say: "... if the area is affected by a mesoscale meteorological feature that is simulated incorrectly by the model, such as a thunderstorm, valley breeze, or sea breeze, the model will be similarly deficient. Therefore, we do not recommend using the downscaling technique in areas where the emission inventory or meteorology is very uncertain."

**Anonymous Referee #2**

The paper by Goldberg et al. provides an interesting study of using high-resolution CMAQ, vertical profile observations and data sampling techniques to better estimate NO2 VCDs at small scales. The paper is well written, and I have a few suggestions below.

\*Recent studies have revealed NO2 retrieval uncertainties related to structural errors (Lorente et al., 2017 and references therein), including treatments of clouds and aerosols (Lin et al., 2015). These errors are relevant to explanations of errors even in OMI\_CMAQ\_OD. A review of such works is necessary.

**This has been addressed in the paragraph added to the end of the Introduction. We've also now included a table (Table 1), which compares all OMI $NO_2$ retrievals in the literature.**

\*The spatial and temporal matching between CMAQ and OMI is discussed in many places, and sometimes there appears inconsistency [For example, Sect. 2.1 says 'The satellite product was oversampled for June & July over a 5-year period (2008-2012) by re-gridding to the CMAQ 1.33 km model grid and then averaging the data over the 10-month (two months × five years) period.', but Sect. 2.4 says 'To ensure a fair comparison, we average model information to the pixel size.'] Please provide a paragraph in the method section dedicated to data mapping/sampling, including proportioning of pixelbased SWs to CMAQ grid, and refer to this section when mentioning in later sections.

**We have added a new section (Section 2.5), in which this is clarified.**

\*Please describe the model setup (e.g., soil and lightning emissions, vertical layers, model PBL scheme, convection, upper boundary) in Sect. 2.4. This will much help understand the model vertical profiles. The missing soil emissions are not discussed until the line (P14, L1) embedded in Section 3.5.

**All of these have now been included in Section 2.4**

\*Can you compare CMAQ and GMI lightning emissions? I wonder how much of the vertical profile differences are due to lightning (convection) parameterization rather than due to resolution.

We have now added a figure (Figure 3) comparing the lightning NO emissions from the two models. Although GMI simulates larger free tropospheric  $NO_2$ , lightning NO emissions are smaller in GMI. Therefore the differences in free tropospheric  $NO_2$  must arise from an alternative mechanism. Further analysis is beyond the scope of this paper. The end of Section 3.1 is re-phrased as such:

"To determine whether lightning NO is the primary driver of this difference, we compare lightning NO emissions from both model simulations in Figure 3. The CMAQ model ingests lightning NO emissions that are an order of magnitude larger than the GMI simulation at most altitudes. This is likely due to WRF simulating more convective precipitation and higher cloud-top heights, both input variables to the lightning NO parameterization, than GMI. Therefore, the differences in free tropospheric NO2 between the two models likely arise from a combination of the chemistry, aviation emissions, vertical mixing, long-range transport, and stratospheric-tropospheric exchange." \*That model profiles in June/July 2011 are applied to all years needs to be described more clearly in Sect. 2.4. The writing is vague at its current form. Some of the writing on relevant method in the first paragraph of Sect. 3.3.1 should be included in Sect. 2.4. The uncertainty due to interannual variability needs to be discussed.

This has been clarified in our new section (Section 2.5) to state:

*"For years other than 2011, we used 2011 monthly mean values of NO2, temperature, and tropopause pressures for the calculation of the AMF."*

\*At the end of 'Introduction', a summary paragraph showing the novelty of the present study will be very useful.

This has now been included at the end of the Introduction. We have also now included a new table (Table 1), which succinctly describes our study in relation to the previous studies.

At the end of the Introduction, we now state: "We build upon these studies by using an even higher resolution regional air quality model (1.33 km) to generate air mass factors. We use the mid-Atlantic region in the eastern United States as a case study in developing high resolution NO2 tropospheric columns for urban metropolitan areas. Furthermore we utilize a technique for constraining the NO2 shape profiles to aircraft observations and invoke a new downscaling method developed by Kim et al., (2016)."

\*P3, L17 – NO2 is a weak absorber.

**This has been modified to say: "NO2 has strong absorption features within the 400 – 450 nm wavelength region..."**

\*P5, L1 – POMINO does not just provide a higher-resolution retrieval, but it also includes various improvements such as explicit treatment of aerosols and re-calculation of cloud parameters.

This has now been included in the Introduction. We have also now included a new table (Table 1), which describes POMINO in relation to our study."

\*P5, L19-20 – SCD represents light path from the sun to surface/atmosphere and to the instrument.

This has been modified as suggested.

\*P5, L25 – the effects of aerosols are also important.

Aerosol optical depth has been added here.

\*P8, L34 – how to determine the 'best estimate'. I appears that if a 65% overestimate is assumed, the OMI\_GMI result would be closer to EPA values.

The "best estimate" was determined based on the Dunlea et al. 2007 study referenced in Section 2.2.4, which suggests actual  $NO_2$  is 22% lower than chemiluminescence measurements in an urban environment . Lamsal et al., 2008 suggests this number can be up to 65%. Thus we include a range of 3.7 – 10.5 ppb. The CMAQ estimate is within this range, but the GMI estimate is not.

**This has been clarified in the text to state: "... the corrected surface $NO_2$ mixing ratio is approximately 22% lower (but may be up to 65% lower) than observed $NO_2^{*"}$**

\*P11, L9 – should be 'consistently larger'

**Thank you for catching this. It has been modified.**

\*P12, L19-26 – much of the discussion on 'rural' and 'urban' definitions here applies also to discussion in previous sections (i.e., Sect. 3.1) on these environments.

**This paragraph has been shortened and moved to the Discussion section.**

References: Lorente, A., Folkert Boersma, K., Yu, H., Dörner, S., Hilboll, A., Richter, A., Liu, M.-Y., Lamsal, L. N., Barkley, M., De Smedt, I., Van Roozendael, M., Wang, Y., Wagner, T., Beirle, S., Lin, J.-T., Krotkov, N., Stammes, P., Wang, P., Eskes, H. J., and Krol, M.: Structural uncertainty in air mass factor calculation for NO2 and HCHO satellite retrievals, Atmospheric Measurement Techniques, 10, 759-782, doi:10.5194/amt- 10-759-2017, 2017.

**This reference has been added to the text.**

---

## Author Response (AR2)

Reviewer 2:

1. There are several mistakes in the descriptions of NO2 products in Table 1. Taking POMINO as an example, please refer to Table 1 of Lin et al. (2015, ACP) for correct descriptions.

We have clarified the radiative transfer model used for the POMINO product (LIDOVRT v3.6) and clarified that OMLER surface reflectivity was used over the ocean.

2. Treatment of clouds and use of look-up table are both key sources of errors in satellite products. Please specify the cloud dataset and whether a LUT is used in these products.

The OMI $O_2$-$O_2$ cloud pressure/fraction algorithm (Acarreta et al., 2004) was used to determine the location and amount of clouds. There is indeed a look-up table used by this algorithm. Both are clarified in the text.

3. Page 6, Line 17 -- In this study, surface albedo dataset is used rather than surface reflection, which is a limitation. There are many places in the text where "surface reflectivity" should be changed to "surface albedo", the quantity actually used in this study. Also, in this study, aerosol optical depth is not used explicitly to derive the AMF; please revise Line 17.

Surface reflectivity has been changed to surface albedo in all five instances.

While aerosol optical depth was not explicitly used, the scattering weight is implicitly corrected for aerosols within the $O_2$-$O_2$ cloud product. We removed the phrase "aerosol optical depth" and replaced with "aerosols".

**A HIGH-RESOLUTION AND OBSERVATIONALLY CONSTRAINED OMI NO₂ SATELLITE RETRIEVAL**

**Daniel L. Goldberg\*[,1,2], Lok N. Lamsal [3,4], Christopher P. Loughner [5,6], William H. Swartz[4,7], Zifeng Lu [1,2], and David G. Streets [1,2]**

**[1]Energy Systems Division, Argonne National Laboratory, Argonne, IL 60439, USA**

**[2]Computation Institute, University of Chicago, Chicago, IL 60637, USA**

**[3]Goddard Earth Sciences Technology and Research, Universities Space Research Association, Columbia, MD 21046, USA**

**[4]NASA Goddard Space Flight Center, Code 614, Greenbelt, MD 20771, USA**

**[5]NOAA Air Resources Laboratory, College Park, MD 20740, USA**

**[6]Earth System Science Interdisciplinary Center, University of Maryland, College Park, MD 20740, USA**

**[7]Johns Hopkins University Applied Physics Laboratory, Laurel, MD 20723, USA**

**Paper submitted to**

**Atmospheric Chemistry and Physics**

**Originally submitted: March 9, 2017**
**Revised draft submitted:  August 21, 2017**

The submitted manuscript has been created by UChicago Argonne, LLC, Operator of Argonne National Laboratory ("Argonne"). Argonne, a U.S. Department of Energy Office of Science laboratory, is operated under Contract No. DE-AC02-06CH11357. The U.S. Government retains for itself, and others acting on its behalf, a paid-up nonexclusive, irrevocable worldwide license in said article to reproduce, prepare derivative works, distribute copies to the public, and perform publicly and display publicly, by or on behalf of the Government.

[revised manuscript text omitted]